# Genetic regions affecting the replication and pathogenicity of dengue virus type 2

Yoshihiro Samune[1], Akatsuki Saito[2], Tadahiro Sasaki[1], Ritsuko Koketsu[1], Narinee Srimark[3], Juthamas Phadungsombat[1], Masaru Yokoyama[4], Osamu Kotani[4], Hironori Sato[4], Atsushi Yamanaka[3], Saori Haga[5], Toru Okamoto[5], Takeshi Kurosu[6], Emi E. Nakayama[1], Tatsuo Shioda[1]*

1 Department of Viral Infections, Research Institute for Microbial Diseases, Osaka University, Osaka, Japan, 2 Department of Veterinary Science, Faculty of Agriculture, University of Miyazaki, Miyazaki, Japan, 3 Thailand-Japan Research Collaboration Center on Emerging and Re-emerging Infections, Research Institute for Microbial Diseases, Osaka University, Osaka, Japan, 4 Pathogen Genomics Center, National Institute of Infectious Diseases, Musashimurayama, Japan, 5 Institute for Advanced Co-Creation Studies, Research Institute for Microbial Diseases, Osaka University, Osaka, Japan, 6 Department of Virology I, National Institute of Infectious Diseases, Musashimurayama, Japan

* shioda@biken.osaka-u.ac.jp

**Data Availability Statement:** All the information required to reanalyze the data reported in this paper can be found in the manuscript and supporting information.

## Abstract

Dengue is a mosquito-borne disease that has spread to over 100 countries. Its symptoms vary from the relatively mild acute febrile illness called dengue fever to the much more severe dengue shock syndrome. Dengue is caused by dengue virus (DENV), which belongs to the *Flavivirus* genus of the family Flaviviridae. There are four serotypes of DENV, *i.e.*, DENV1 to DENV4, and each serotype is divided into distinct genotypes. Thailand is an endemic area where all four serotypes of DENV co-circulate. Genome sequencing of the DENV2 that was isolated in Thailand in 2016 and 2017 revealed the emergence of the Cosmopolitan genotype and its co-circulation with the Asian-I genotype. However, it was unclear whether different genotypes have different levels of viral replication and pathogenicity. Focus-forming assay (FFA) results showed that clinical isolates of these genotypes differed in focus size and proliferative capacity. Using circular polymerase extension reaction, we generated parental and chimeric viruses with swapped genes between these two DENV2 genotypes, and compared their focus sizes and infectivity titers using FFA. The results showed that the focus size was larger when the structural proteins and/or non-structural NS1-NS2B proteins were derived from the Cosmopolitan virus. The infectious titers were consistent with the focus sizes. Single-round infectious particle assay results confirmed that chimeric viruses with Cosmopolitan type structural proteins, particularly prM/E, had significantly increased luciferase activity. Replicon assay results showed that Cosmopolitan NS1-NS2B proteins had increased reporter gene expression levels. Furthermore, in interferon-receptor knock-out mice, viruses with Cosmopolitan structural and NS1-NS2B proteins had higher titers in the blood, and caused critical disease courses. These results suggested that differences in the sequences within the structural and NS1-NS2B proteins may be responsible for the differences in replication, pathogenicity, and infectivity between the Asian-I and Cosmopolitan viruses.

**Funding:** E.E.N.:Japan Agency for Medical Research and Development (AMED; grant number: 22wm0125010h0003). https://www.amed.go.jp/en/index.html Y.S.:Japan Science and Technology Agency Support for Pioneering Research Initiated by the Next Generation (JST SPRING; grant number: JPMJSP2138). https://www.jst.go.jp/EN/ These funders played no role in the study design, data collection and analysis, decision to publish, or preparation of the manuscript.

**Competing interests:** The authors have declared that no competing interests exist.

## Author summary

Since the discovery of dengue virus (DENV) in 1907, many people in Latin America, and South and Southeast Asia have been at risk of infection. Analysis of the sequences of viruses isolated in Thailand in 2016 and 2017 revealed the emergence of the Cosmopolitan genotype of DENV type 2 (DENV2), and its co-circulation with the pre-existing Asian-I genotype of DENV2. This co-circulation still exists in Thailand. We noted that there were large differences in the infectious titers between these genotypes, and the focus sizes also differed. In the present study, we analyzed both the *in vitro* and *in vivo* characteristics of these viruses using multiple recombinant viruses to determine which viral genetic region is responsible for these differences. We identified a genetic region encoding structural and NS1-NS2B proteins that affects the infectivity, replication, and pathogenicity of the viruses. Our study provides new insights into the transmission of new DENVs, and the interactions between DENV proteins.

## Introduction

Dengue fever is an infectious disease caused by dengue virus (DENV), which is transmitted by *Aedes aegypti* and *Aedes albopictus* mosquitoes. The virus infects approximately 400 million people each year, and causes symptoms in approximately 100 million people [1]. In addition, an estimated 250,000 people develop severe dengue hemorrhagic fever. Outbreaks have been confirmed in Southeast Asia, South Asia, Central and South America, the Caribbean islands, and Africa, where the vector mosquitoes are present. In Japan, an outbreak was confirmed in Tokyo in 2014 [2]. Currently, there are no direct-acting drugs against the virus, and treatments are available only for treating the symptoms [1,3,4].

DENV belongs to the genus *Flavivirus*. The viral genome is a single-stranded RNA of approximately 11 kb in length that encodes three structural proteins, *i.e.*, capsid (C), precursor membrane (prM) and envelope (E) proteins, and seven non-structural proteins, *i.e.*, NS1, NS2A, NS2B, NS3, NS4A, NS4B and NS5, which are encoded in this order from the 5' end. The 5' end has a cap structure, and the 3' end is not polyadenylated [1,5]. The structural proteins form viral particles while the non-structural proteins function in host immune evasion as well as virus replication in infected cells [5].

DENV consists of four serotypes (DENV1, DENV2, DENV3, and DENV4), and each serotype is further divided into several genotypes. Five genotypes of DENV2 are known: Cosmopolitan, Asian-I, Asian-II, Asian/American, and American [6–8]. In Thailand, where DENV2 was isolated for this study, all four serotypes of DENV coexist, and cases of dengue fever are confirmed every year [9].

We previously analyzed 21 DENV strains isolated in 2016 and 2017 in Thailand; eight of these strains were DENV2. Genome sequencing of these DENV2 isolates revealed that the Cosmopolitan type coexisted with the Asian-I type, which was previously dominant in Thailand [10]. These genotypes were still co-existing in 2020 [9].

In the present study, we analyzed the biological characteristics of these DENV2 clinical isolates, and found differences in the focus sizes on the focus-forming assay (FFA) and in the proliferative potential in tissue cultures. We describe herein the identification of the genetic regions responsible for the differences in virus replication, pathogenicity, and infectivity between the genotypes Cosmopolitan and Asian-I.

## Materials and methods

### Ethics statement

The present study was approved by the Animal Experiment Committee of the Research Institute for Microbial Diseases, Osaka University (R01-08-0).

### Cell cultures

Vero cells (ATCC CCL-81) were cultured in minimum essential medium (MEM; 21442–25; Nacalai Tesque) supplemented with 10% fetal bovine serum (FBS), 100 U/mL penicillin and 100 μg/mL streptomycin (Pe/St; 26253–84; Nacalai Tesque), and 1% Non-Essential Amino Acid (11140–050; Nacalai Tesque) in an atmosphere with 5% $CO_2$. BHK-21 cells (ATCC CCL-10) were cultured in MEM supplemented with 5% FBS and Pe/St. Lenti-X 293T cells (ATCC CRL-3216) were cultured in Dulbecco's modified Eagle's medium (DMEM; 08458–16; Nacalai Tesque) supplemented with 10% FBS and Pe/St. C6/36 cells (ATCC CRL-1660) were cultured in Leibovitz's (1×; 11415–064; Gibco) supplemented with 0.3% tryptose phosphate broth powder, 10% FBS, and Pe/St at 28˚C without $CO_2$.

### Generation of recombinant viruses

Recombinant viruses were generated by the circular polymerase extension reaction (CPER) method as reported previously [11]. The whole genomes of the Th16-005DV2 and Th16-026DV2 strains were amplified by RT-PCR using PrimeSTAR GXL DNA Polymerase (R050A; TaKaRa) to generate five fragments. The primer pairs used were designed to have a 25-nucleotide overlap at the end of each fragment (S1 Table). The PCR fragments were cloned into TOPO TA Cloning Vector pCR2.1 (45–0641; Thermo Fisher Scientific). In addition, DNA fragments encoding a polyadenylation (A) signal, hepatitis delta virus ribozyme (HDVr), and cytomegalovirus (CMV) promoter were synthesized and cloned into the pCR-TOPO vector to produce the pCR-TOPO-pCMV-RZM plasmid. The chimeric fragments among the S, prM, and E genes of fragment 1, and among the NS1, NS2A, and NS2B genes of fragment 2 were generated by the overlapping PCR method using the primers listed in S1 Table. Five DENV-coding fragments and linker fragments were prepared by PCR from cloning vectors, the sequence authenticities of the cloned fragments were confirmed, and the CPER reaction (denaturation at 98˚C for 2 min, 20 cycles of extension at 98˚C for 10 sec, 55˚C for 15 sec and 68˚C for 12 min, and a final extension at 68˚C for 12 min) was performed using PrimeSTAR GXL DNA Polymerase to obtain circular DNA. The CPER product was then transfected into BHK-21 cells using TransIT-LT1 Transfection Reagent (MIR2300; TaKaRa). At day 6 post-transfection, the culture supernatant was collected and treated with 0.1 U/mL TURBO DNase (AM2238; Thermo Fisher Scientific) for 1 h at 37˚C to remove the remaining plasmid DNA. Viruses were propagated in C6/36 cells, and the levels of viral RNA were measured with a real-time RT-PCR assay to normalize the input viruses for further experiments.

### Quantification of viral RNA

Viral RNA was extracted from 140 μL of culture supernatant using the QIAamp Viral RNA Mini Kit (52926; QIAGEN) according to the manufacturer's protocol. The One-Step SYBR PrimeScript RT-PCR Kit II (RR086A; TaKaRa) and universal primers for DENV (DN-F and DN-R; S1 Table) were used for quantification [12]. The total reaction volume was 12.5 μL/tube, and the PCR conditions were: reverse transcription at 42˚C for 5 min and 95˚C for 10 min, followed by 40 cycles of 95˚C for 5 sec and 60˚C for 34 sec. The laboratory DENV2 strain 16681 (NC_001474.2) with known focus-forming units (FFU) was used as a reference.

Fluorescence signals were measured using an Applied Biosystems 7500 Fast Real-Time PCR System and QuantStudio3 (Thermo Fisher Scientific). The levels of viral RNA are expressed as the copy numbers relative to that of the reference strain 16681 with the assumption that 1 FFU of the strain 16681 contained one copy of viral RNA.

## Infection

Vero cells were plated at $4.2 \times 10^4$ cells/well in a 24-well plate, and incubated overnight prior to infection. The following day, the cells were infected with DENV at RNA copy numbers equivalent to a multiplicity of infection (MOI) of 0.5 FFU/cell. Stock viruses were diluted with MEM without FBS to equalize the concentration in all samples. After 2 h of infection, the cells were washed with MEM containing 2% FBS, then 500 μL of fresh medium was added. The culture supernatant was collected every 24 h, and the viral RNA was measured with a real-time RT-PCR assay.

## Focus-forming assay

Vero cells were plated at $1.0 \times 10^4$ cells/well or $4.0 \times 10^4$ cells/well in a 96-well plate, and incubated overnight prior to infection. The following day, the cells were infected with DENV for 2 h. The viruses were serially diluted with MEM without FBS. After the 2 h of infection, 100 μL of a mixture of MEM (2×; 11935–046; Gibco), 3% FBS, and 3% carboxymethyl cellulose was added to each well without removing the viral solution, and the cells were incubated at 37°C for 72 h. Then, the cells were washed with PBS, and fixed with 4% formalin in PBS for 30 min at room temperature. Next, 4G2 monoclonal antibody against pan-DENV E protein (ATCC HB-197) was added, and the cells were incubated at 37°C for 1 h. Subsequently, the cells were washed with PBS, then incubated with 3000-fold-diluted (0.3 μg/mL) horseradish peroxidase-labeled goat polyclonal anti-mouse IgG (H+L) (5220–0341; KPL) at 37°C for 1 h. Finally, the cells were washed with PBS, and a reaction mixture containing Metal Enhancer for DAB Stain (07388–24; Nacalai Tesque) plus Peroxidase Stain DAB Kit (Brown Strain; 25985–50; Nacalai Tesque) reagents was added. After incubation for 2 to 10 min at room temperature, the plate was washed with water, then dried, and the foci were counted using an ELISPOT reader (ImmunoSpot S6 VERSA Analyzer; Cellular Technology). The focus sizes were also measured using the ELISPOT reader. The focus size range to be measured was set to 0.0000 to 9.6296 mm$^2$, and the diffuseness was set to Largest+1.

## Single-round infectious particles (SRIPs) assay

Lenti-X 293T cells were plated at $5 \times 10^5$ cells/well in a 6-well plate, and incubated overnight prior to transfection. The cells were transfected with a combination of three plasmids: 0.5 μg of the prME expression plasmid, 0.5 μg of the C expression plasmid, and 1.0 μg of the reporter plasmid [13]. These plasmids were mixed in OPTI-MEM (31985–062; Gibco), and transfected using TransIT-293 Transfection Reagent (MIR2700; TaKaRa). The supernatant was collected at 3 days post-transfection, and treated with TURBO DNase (AM2238; TaKaRa). The titer of the SRIPs was determined by real-time RT-PCR using primers DN-F and DN-R (S1 Table). Vero cells were plated at $8.3 \times 10^4$ cells/well in a 24-well plate, and incubated overnight prior to infection. The next day, the culture medium was removed from the plate, and the cells were infected with diluted SRIPs for 2 h. The SRIPs were diluted with MEM containing 2% FBS to equalize the concentration in all samples. After infection, the cells were washed twice with MEM containing 2% FBS, then 100 μL of fresh MEM containing 2% FBS was added, and the cells were incubated at 37°C for 72 h. Subsequently, the wells were washed with PBS, and

treated with 1× Passive Lysis Buffer (E194A; Promega) for 10 min. Luciferase activity was measured using the Nano-Glo Luciferase Assay System (N1120; Promega).

## Replicon assay

Lenti-X 293T cells were plated at $2.5 \times 10^5$ cells/well in a 12-well plate, and incubated overnight prior to transfection. The cells were transfected with the CPER product. The culture supernatant was collected every 24 h after transfection. Luciferase activity was measured using the Secrete-Pair Luminescence Assay Kit (LF062; GeneCopoeia)

## Infection of IFNα/β and IFNγ receptor double knockout (IFNR-KO) mice

We used 4-week-old IFNR-KO mice, which have been described previously [14]. Virus solution diluted in PBS to equalize the amount of RNA in all samples was administered intraperitoneally. After infection, tail vein or heart blood sampling was performed at regular intervals. The body weight was measured using an animal scale (FX-2000i; A&D Company), and the mice were euthanized when the predetermined humane endpoint was reached. Rearing and experiments were conducted in accordance with the regulations of the Animal Experiment Committee of the Research Institute for Microbial Diseases, Osaka University. The cages were maintained at 20˚C to 24˚C.

## Molecular modeling

As templates for homology modeling, we first obtained the 3.5 Å resolution cryo-electron microscopy structure of the mature DENV E protein (PDB ID: 3J27), and the 2.2 Å resolution crystal structure of the DENV prM-E protein complex (PDB ID: 3C5X) from the RCSB Protein Data Bank. Then, the sequence alignments of the targets, *i.e*., Th16-005DV2 (Asian-I) or Th16-026DV2 (Cosmopolitan) E protein, to the templates were generated using MOE-Align in the Molecular Operating Environment (MOE) version 2020.0901 (Chemical Computing Group, Inc., Quebec, Canada). Next, three-dimensional (3D) models of the two neighboring sets of the mature E and M protein hetero-tetramer or the two neighboring sets of the prM-E protein hetero-tetramer in DENV of Th16-005DV2 or Th16-026DV2 were constructed by the homology modeling technique using MOE-Homology in MOE as described previously [15,16]. For each homology modeling in MOE, we obtained 25 intermediate models. We selected the 3D intermediate models with the best scores according to the generalized Born/ volume integral method [17]. The 3D structures were thermodynamically optimized by energy minimization using MOE and an Amber 10:extended Hückel theory force field. The force field was combined with a parameter for proteins and nucleic acids [18], and small molecules using the two-dimensional extended Hückel theory [19].

## Quantification and statistical analysis

GraphPad Prism 8 was used for data analyses and visualization. The specific statistical tests and the numbers of animals included are described in the figure legends.

## Results

### Growth characteristics of the clinical strains of DENV2

The FFA results for six strains of DENV2 (three Asian-I and three Cosmopolitan strains), which were previously isolated by Phadungsombat *et al.* [10], revealed that the focus sizes varied among the strains: the Cosmopolitan strains had larger foci than the Asian-I strains (Fig 1A). In addition, the Cosmopolitan strains proliferated faster than the Asian-I strains in both

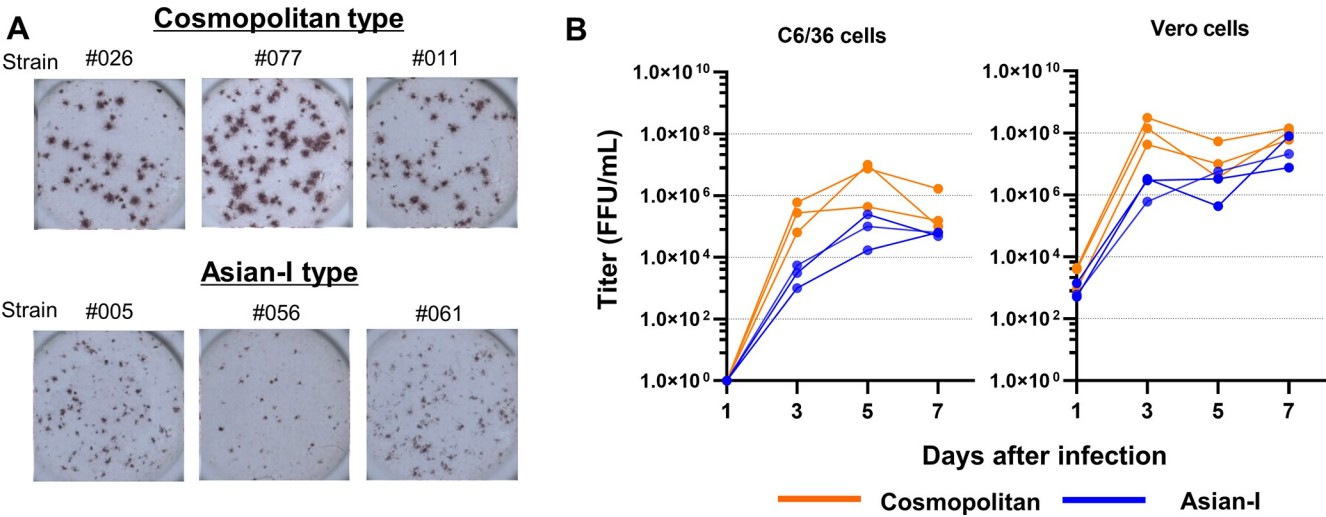

**Fig 1. Characteristics of the clinical isolates.** (A) Images of the foci of the co-circulating clinical isolates of dengue virus type 2 (DENV2) in Thailand. Cosmopolitan type clinical strains Th16-026DV2 (026), Th17-077DV2 (077) and Th16-011DV2 (011), and Asian-I type strains Th16-005DV2 (005), Th16-056DV2 (056), and Th17-061DV2 (061) were used. (B) The proliferation rate of the six clinical isolates in C6/36 and Vero cells. Viruses were inoculated into cells at a multiplicity of infection of 0.1 per cell, and culture supernatants were periodically assayed for the levels of infectious virus by determining the focus-forming units (FFU) in Vero cells.

mosquito C6/36 cells and mammalian Vero cells (Fig 1B). We selected one strain of each genotype, Th16-005DV2 (NCBI Accession No. LC410184) and Th16-026DV2 (NCBI Accession No. LC410190), which were isolated in the same year (2016), for further experiments, as described below.

## Generation of DENVs by the CPER method

Setoh *et al.* [11] established the CPER method for generating new Flavivirus Zika viruses from its cDNA. In their method, several protein-coding regions are split into different fragments of cDNA (Fig 2A). To facilitate the exchange of protein-coding regions between different genotypes of DENV2, we re-designed the primers to produce five viral genome fragments that are divided at the partition of each protein-coding region (Fig 2A). The five fragments included one fragment covering the structural proteins C, prM and E, and four fragments covering the non-structural proteins NS1-NS2B, NS3, NS4A-NS4B, and NS5.

The CPER products were transfected into BHK-21 cells, and the culture supernatants were collected. The amount of DENV2 RNA in each culture supernatant was subsequently measured by real-time RT-PCR (Fig 2B). The RNA of DENV2 was detected in the CPER samples, and the levels were very low in the negative-control samples of mutant NS5-C709A with inactivated RNA polymerase for both the Cosmopolitan and Asian-I types [20]. The level of DENV2 RNA was higher with Cosmopolitan Th16-026DV2 than with Asian-I Th16-005DV2. The culture supernatants of the cells transfected with CPER products were then inoculated into C6/36 cells to propagate the resultant viruses after equalization of the DENV2 RNA levels between the Cosmopolitan and Asian-I viruses by diluting the culture supernatants. After 6 days of infection, the amount of DENV2 RNA in the culture supernatants was measured by real-time RT-PCR (Fig 2C). The results showed that the amount of DENV2 RNA was higher with the Cosmopolitan virus than with the Asian-I virus. In the negative-control samples, virus did not grow at all even when all of the transfection supernatant was used to inoculate the C6/36 cells (Fig 2C).

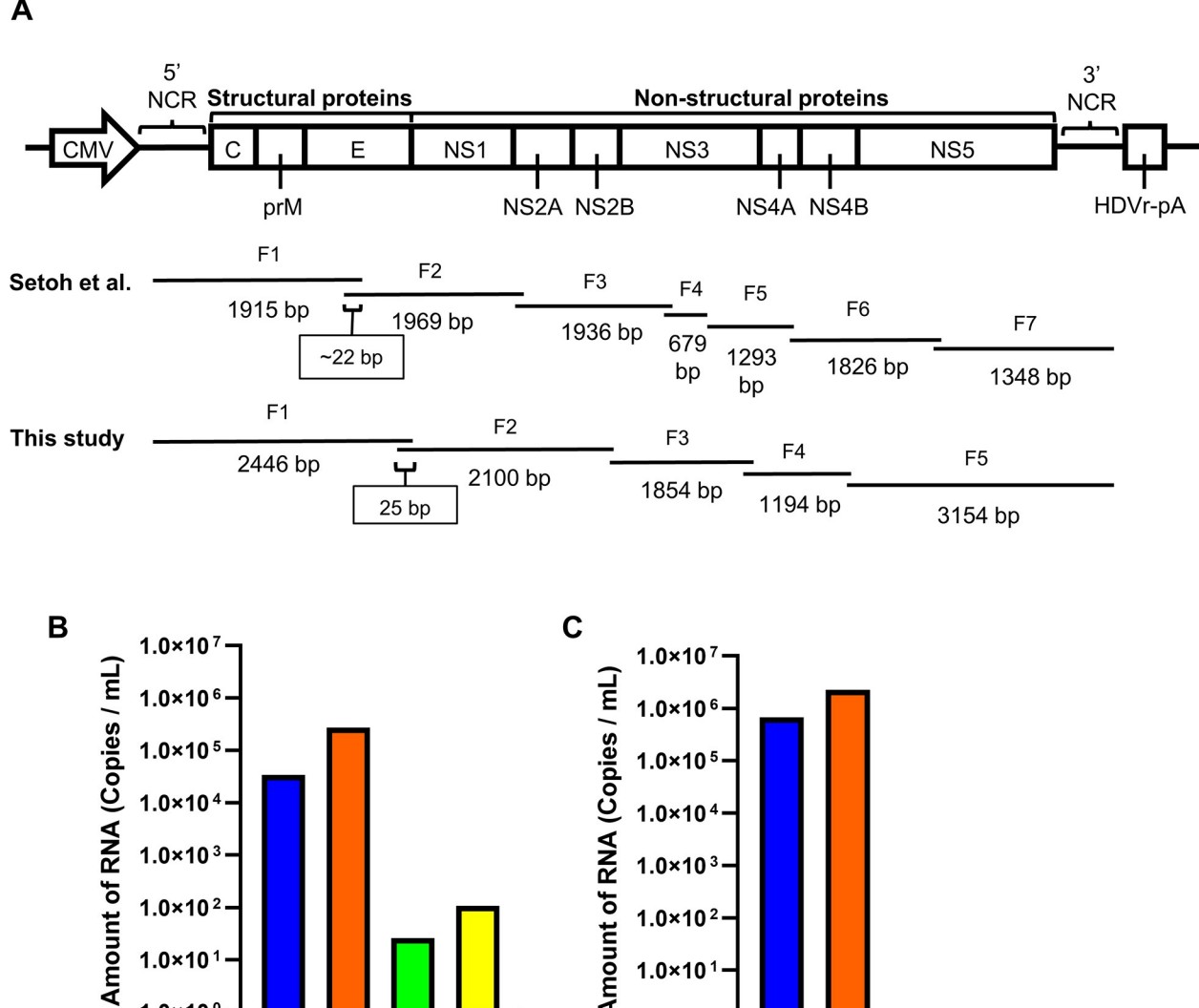

**Fig 2. Generation of DENV2 by the circular polymerase extension reaction (CPER) method.** (A) Schematic representation of the fragmentation of the whole dengue virus genome for the CPER method. In a previous publication by Setoh *et al*. (mSphere.2017; 2(3)) seven fragments were designed with an overlapping region of about 22 nt at the end. In the present study, five fragments were used, and each fragment was designed to have a 25-nt-overlapping region at the end. These fragments were mixed with a linker fragment containing the cytomegalovirus (CMV) promoter, hepatitis D virus ribozyme (HDVr), and a poly(A) tail (pA). (B) Comparison of the amounts of viral RNA in the culture supernatants at 6 days after the transfection of CPER products into BHK-21 cells. (C) Amounts of RNA in viruses propagated in C6/36 cells with the culture supernatants of transduced BHK-21 cells. Asian-I and Cosmopolitan viruses were infected at a multiplicity of infection of 0.01 copies/cell. We used NS5-C709A mutants as non-replicating negative controls.

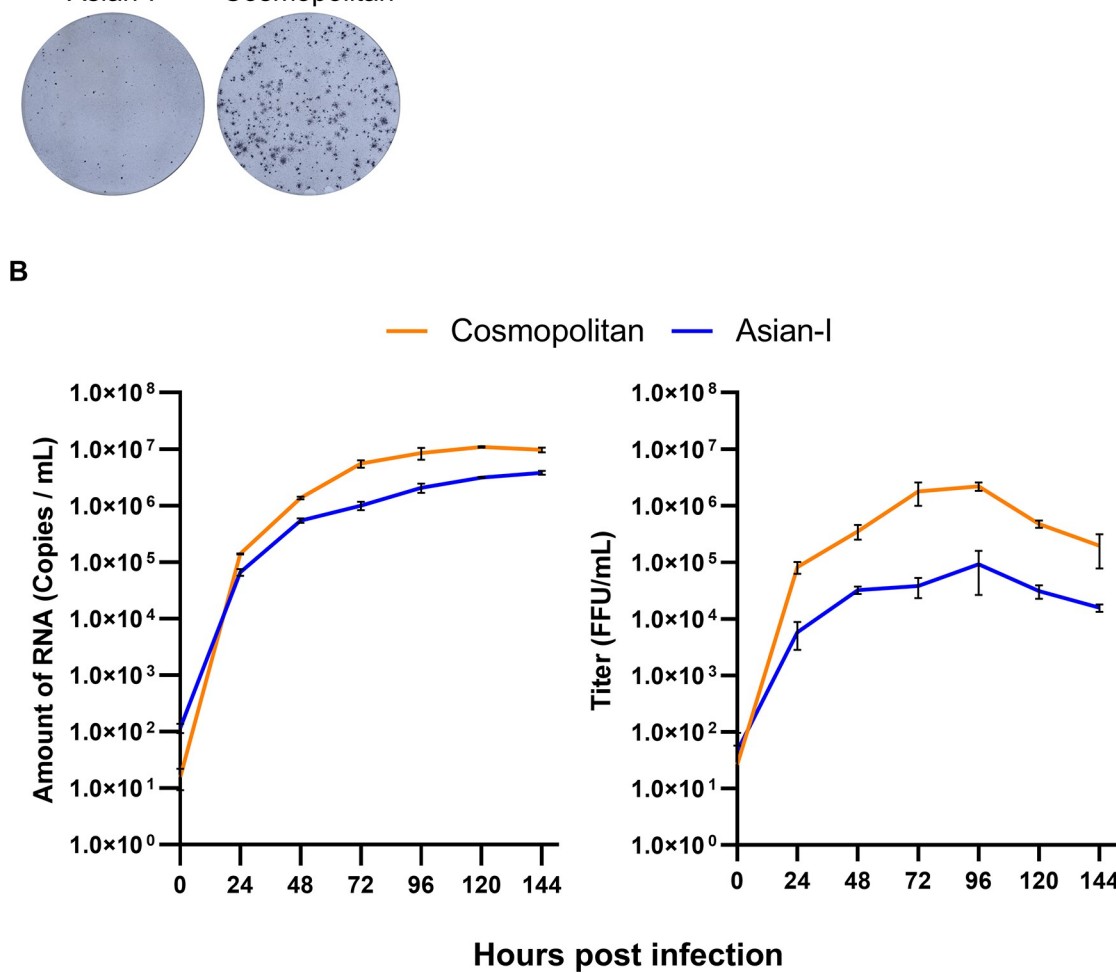

**Fig 3. Characteristics of the DENV2 generated by the CPER method.** (A) Images of the foci of the recombinant DENV2 obtained by the ELISPOT reader. (B) Growth curves of the recombinant viruses in Vero cells infected with virus samples at a multiplicity of infection of 0.5 copies/cell. Culture supernatants were harvested every 24 h, and assayed for the level of viral RNA and the infectious titer. The means and standard deviation of triplicate samples are shown.

Stock recombinant viruses were diluted to equalize the virus concentration, then inoculated into Vero cells at a dose equivalent to a MOI of 0.5 FFU/cell for further focus formation and growth kinetics experiments. The recombinant DENV2 propagated in C6/36 cells reproduced the phenotypes observed in the original clinical isolates, *i.e.*, the recombinant Cosmopolitan virus resulted in larger focus sizes (Fig 3A) and showed faster growth kinetics (Fig 3B) than the recombinant Asian-I virus. While the RNA copy numbers in the supernatant differed by one order of magnitude between the Cosmopolitan and Asian-I types (Fig 3B, left panel), the difference in infectious virus production was more than 20-fold (Fig 3A, right panel). This discordance indicated that the genetic background affected virus infectivity.

## Generation of chimeric DENV2 for the Cosmopolitan and Asian-I genotypes

To identify the genetic regions responsible for the differences in the proliferative and infectious potential between the Asian-I and Cosmopolitan viruses, round-robin chimeric viruses

were generated using the CPER method (Fig 4A). Chimeric viruses containing swapped genes were successfully recovered (S1 Fig), and the levels of viral RNA in the stock viruses were measured with a real-time RT-PCR assay to normalize the input viruses for further experiments.

Switching of the fragments in the background of the Asian-I virus to the Cosmopolitan virus was expected to increase the growth capability and the focus sizes. Vero cells were infected with the chimeric viruses at a MOI of 0.5 FFU/cell, and the infectious titers 3 days after infection are shown in Fig 4B. All six viruses with titers lower than that of the parental Asian-I virus (Fig 4B, blue bars) carried the Asian-I virus structural proteins covered by the F1 fragment, and NS1, NS2A, and NS2B covered by the F2 fragment. The remaining virus carrying the Asian-I structural and NS1, NS2A, and NS2B proteins (AACAA) showed almost the same titers as the parental Asian-I virus. This indicated a correlation between the Asian-I virus structural and NS1, NS2A, and NS2B proteins and low infectious titers. On the other hand, among the eight viruses with titers higher than that of the parental Cosmopolitan virus (Fig 4B, red bars), six viruses carried the Cosmopolitan structural and NS1, NS2A, and NS2B proteins, and two viruses carried the Cosmopolitan structural and Asian-I NS1, NS2A, and NS2B proteins (CACAC and CACCA). The remaining two viruses carrying the Cosmopolitan structural and NS1, NS2A, and NS2B proteins (CCCAC and CCACC) showed high titers similar to the parental Cosmopolitan virus. These results suggested a correlation between the Cosmopolitan structural and NS1, NS2A, and NS2B proteins and high infectious titers (Fig 4B). Furthermore, as shown in Fig 4C, the focus sizes also tended to be larger in viruses carrying the Cosmopolitan structural and NS1, NS2A, and NS2B proteins.

As biological replicates, we repeated the experiments in Figs 3 and 4 using virus stocks prepared independently from the aforementioned stocks on parental and chimeric viruses for the structural and NS1, NS2A, and NS2B proteins, and confirmed our conclusions that differences in the structural and NS1-2B coding sequences accounted for the faster growth kinetics of the Cosmopolitan type virus (S2A and S2B Fig).

Post-hoc sequence analysis of the virus stocks generated above identified one unintended mutation in the CCACC virus that led to a substitution of glutamic acid (E) to lysine (K) at position 202 of E. However, exclusion of this CCACC virus from the analysis did not affect the above conclusions.

## Evaluation of structural proteins

Fig 4 shows that the replacement of structural proteins alone could, at least in part, switch the viral phenotypes. We therefore further examined the structural proteins, C, prM, and E. A series of chimeric viruses, as shown in Fig 5A, were generated and evaluated for their infectious titers and focus sizes. These chimeric viruses carried various combinations of structural proteins from the Asian-I or Cosmopolitan viruses in the same background of Asian-I nonstructural proteins. As shown in Fig 5B, the infectious titers became even lower than that of the parental Asian-I virus (AAA-AAAA) when prM and E were derived from different genotypes (AAC-AAAA, ACA-AAAA, CAC-AAAA, and CCA-AAAA). In addition, the recombinant virus with the Cosmopolitan prM and E proteins (ACC-AAAA) showed only slightly reduced infectious titers with almost comparable focus sizes as those of the virus with the Cosmopolitan structural proteins (CCC-AAAA; Fig 5C and 5D). These results suggest that the interaction between prM and E is important in maintaining proper infectious titers. On the other hand, the effect of C was small, *i.e.*, the titers of the CAA-AAAA virus were comparable to those of the AAA-AAAA virus. Post-hoc sequence analysis of the virus stocks generated above confirmed the sequence authenticity of the viruses. Next, we decided to analyze the structural proteins, prM and E, in more detail.

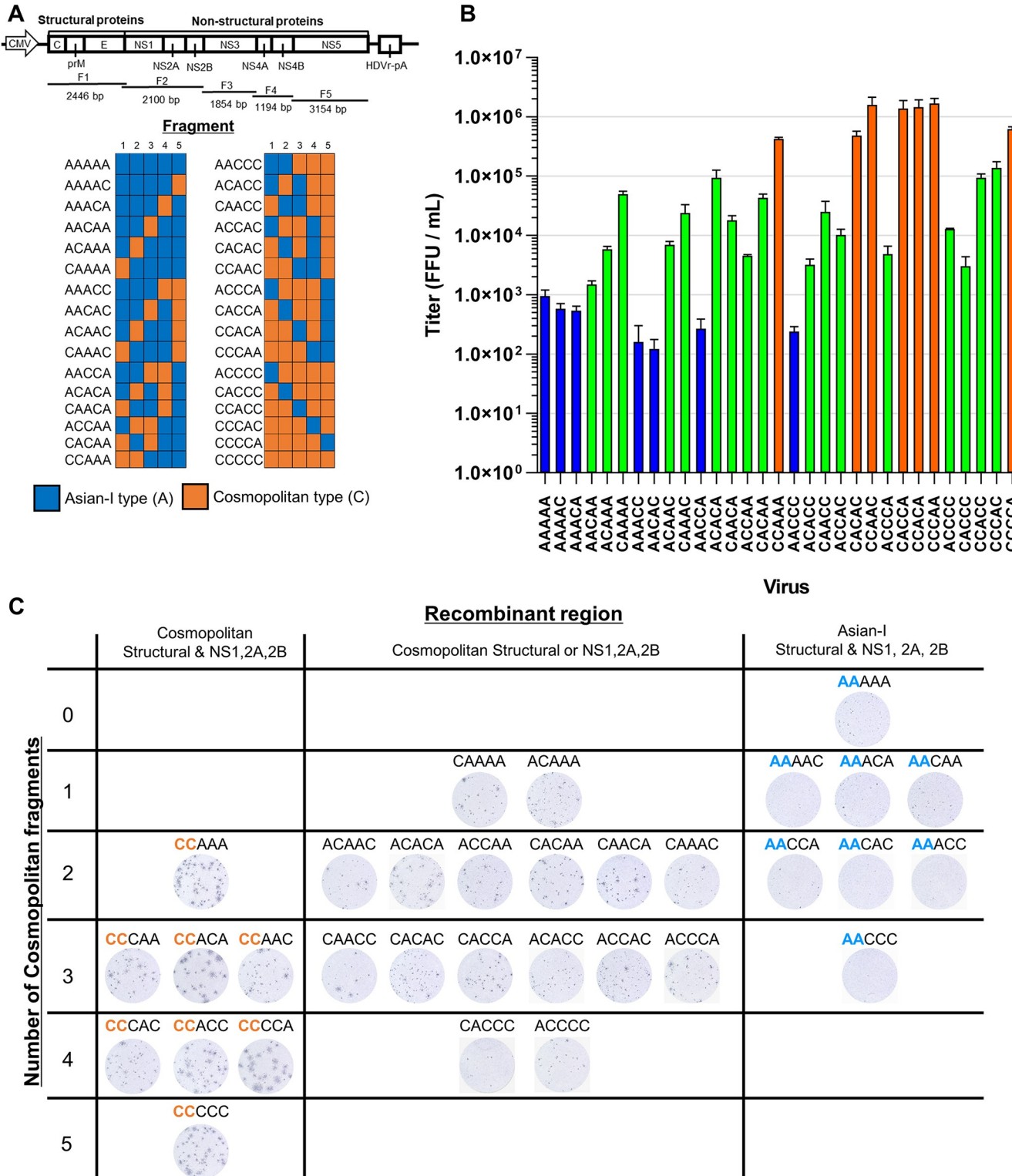

**Fig 4. Characteristics of the chimeric DENV2 between the Cosmopolitan and Asian-I genotypes.** (A) Schematic representation of 30 chimeric viruses with two different parental strains. The fragments derived from the Asian-I [A] genotype and Cosmopolitan [C] genotype are shown in blue and orange, respectively. AAAAA and CCCCC indicate the parental Asian-I and Cosmopolitan viruses, respectively. (B) The FFU of the generated recombinant viruses 3 days after the infection of Vero cells at a multiplicity of infection of 0.5 copies/cell. Viruses with titers equal to or lower than that of the parental Asian-I strain are shown in blue. Viruses with titers equal to or higher than that of the parental Cosmopolitan strain are shown in orange. Viruses with titers in between those

of the two parental viruses are shown in green. The means and standard deviation of triplicate samples are shown. (C) Images of the foci of all recombinant viruses. The chimeric viruses are grouped according to the number of Cosmopolitan virus-derived fragments and whether the structural and NS1, NS2A, and NS2B proteins were derived from the Cosmopolitan or Asian-I viruses. Red CC and blue AA indicate the viruses with structural and NS1, NS2A, and NS2B proteins derived from the Cosmopolitan and Asian-I viruses, respectively.

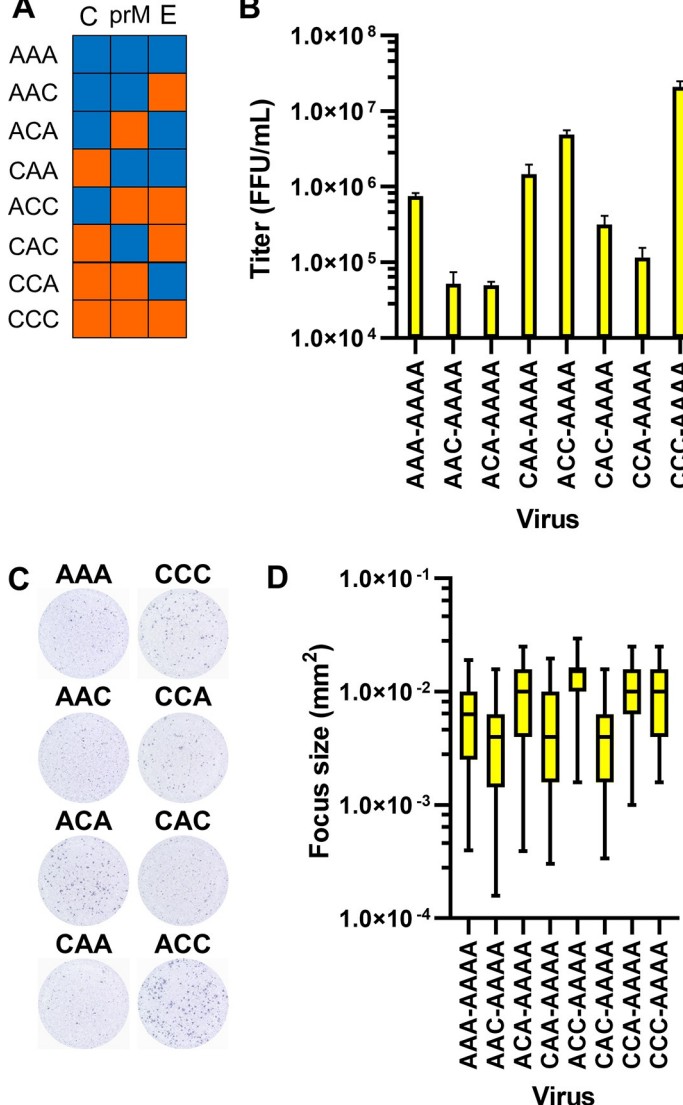

**Fig 5. Characteristics of viruses with recombinant structural proteins.** (A) Schematic representation of six viruses containing chimeric structural proteins (C, prM, and E). The fragments derived from the Asian-I [A] genotype and Cosmopolitan [C] genotype are shown in blue and orange, respectively. AAA and CCC indicate the Asian-I and Cosmopolitan structural proteins, respectively. (B) Infectious titers of the viruses containing chimeric structural proteins 3 days after the infection of Vero cells at a multiplicity of infection of 0.0625 copies/cell. The means and standard deviation of triplicate samples are shown. (C) Representative images of the foci of the chimeric viruses. (D) Box plots of the focus sizes of the chimeric viruses. The center lines in boxes, boxes, and horizontal bars above or below the boxes indicate the median, interquartile, and 5 or 95 percentiles, respectively, from triplicate wells.

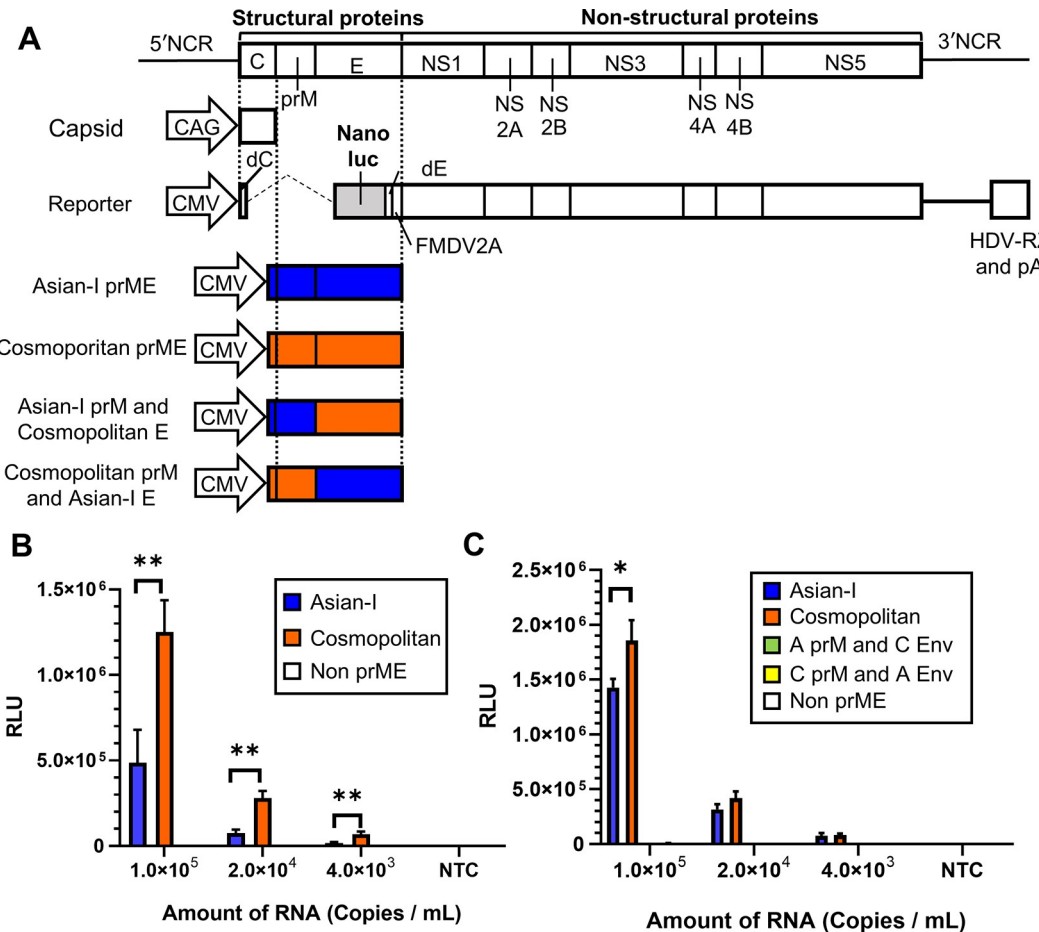

**Fig 6. Evaluation of infectivity by a single-round infectious particle (SRIP) assay.** (A) Schematic representation of the reporter vector showing the position of the cytomegalovirus promoter (CMV), NanoLuc gene, 2A protein sequence of foot-and-mouth disease virus (FMDV2A) for self-excision, hepatitis delta virus ribozyme (HDV-RZ), and polyadenylation signal (pA). The structural protein-expressing plasmids (prME) used to generate SRIPs are also shown. Blue and red indicate the sequences derived from the Asian-I and Cosmopolitan viruses, respectively. (B, C) Luciferase activity of the SRIPs produced by the transfection of Lenti-X 293T cells with the reporter plasmid and structural protein-expressing plasmids as indicated. The multiple t test results with a statistically significant difference are indicated by asterisks (*:P < 0.05, **:P < 0.01). The means and standard deviation of triplicate samples are shown. Representative results of three independent experiments are shown.

## Evaluation of structural proteins by a single-round infectious particles (SRIP) assay

A reporter assay using SRIPs was performed to evaluate the infectivity of the prM or E protein-swapped viruses. SRIPs were produced by the transfection of three different plasmids, *i.e.*, a C-expressing plasmid, a prM/E-expressing plasmid, and a NanoLuc luciferase-expressing reporter plasmid, as shown in Fig 6A. After infection, the SRIPs transiently express luciferase. The SRIPs with Cosmopolitan prM/E showed approximately three times higher luciferase activity than those with Asian-I prM/E after the input viruses were normalized by the amount of viral RNA (Fig 6B). These results confirmed the importance of prM/E in viral infectivity, which was seen in the experiments with the live chimeric viruses (Fig 5). In addition, SRIPs with chimeric prM and E failed to express luciferase activity at all (Fig 6C), which also confirmed the importance of the combination of prM and E, which was seen in the experiments with the live chimeric viruses (Fig 5).

## Evaluation of the viral proliferative potential of non-structural proteins by a replicon assay

To evaluate the effect of differences in non-structural proteins on DENV replication, we performed a replicon assay using the Gaussia luciferase (Gluc) gene as a reporter. A DNA fragment in which the Gluc gene was introduced instead of the structural protein-coding region and the fragments encoding the non-structural proteins were linked by the CPER method. The resultant CPER product was then transfected into cells, and the levels of Gluc in the culture supernatant were measured (Fig 7A). At first, luciferase activity is derived simply from the transfected CPER product, but it increases later due to the Gluc-containing genome replication supported by non-structural proteins [21]. To control for the transfection efficiency

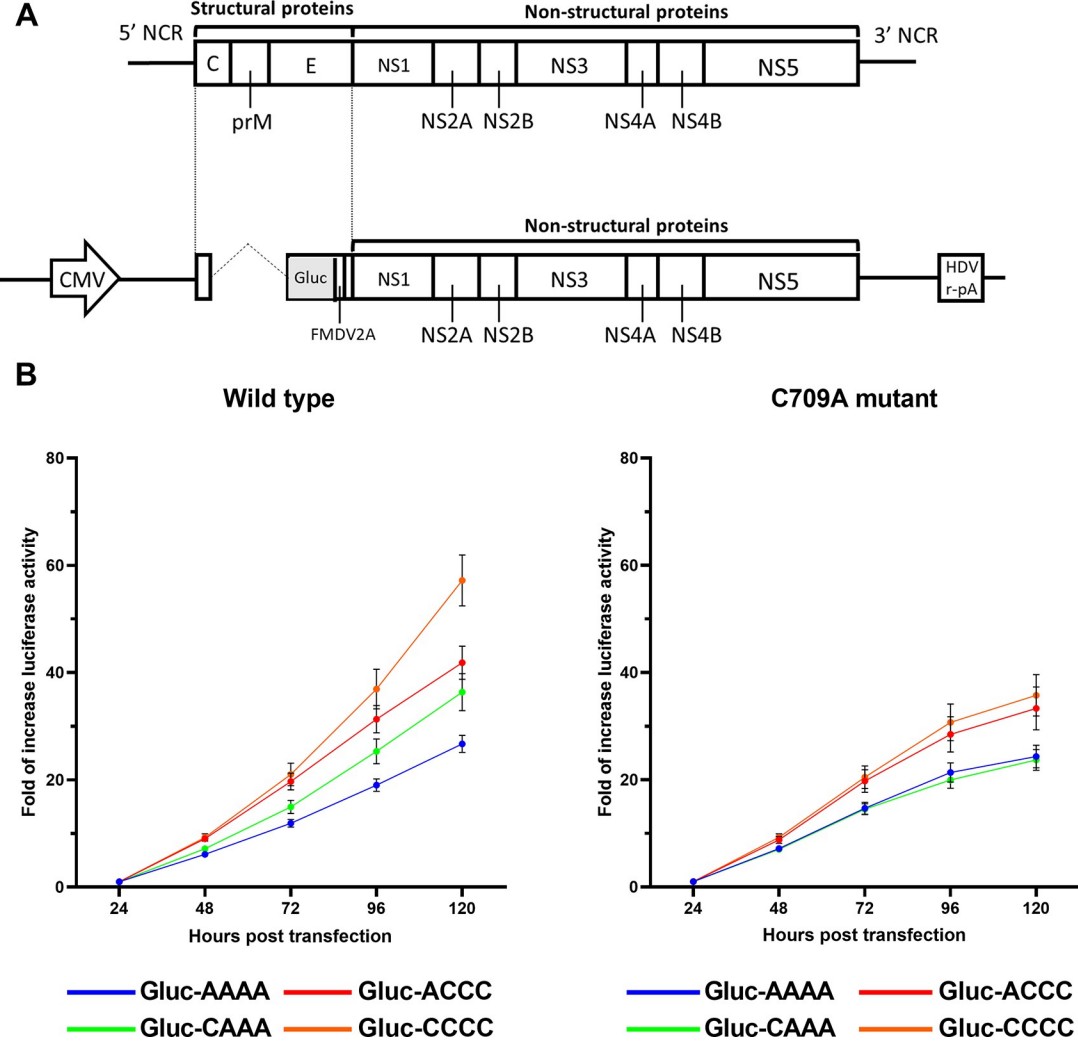

**Fig 7. Evaluation of proliferation by the replicon assay.** (A) Structure of the CPER product used in the replicon assay. Schematic representation of the replicon construct showing the position of the Gaussia luciferase gene (Gluc) and foot-and-mouth disease virus 2A peptide (FMDV2A). (B) The fold increase in luciferase activity when compared to the activity at 24 h after the transfection of DENV2 replicons. Lenti-X 293T cells were transfected with the CPER product, and luciferase activity was monitored at the indicated time points. The means of triplicate samples were calculated in each of the three independent experiments and the averages and standard deviation (SD) of the means of the three independent experiments are shown. Statistical significance of the differences were tested using the Holm-Sidak method with alpha = 0.05. Each row was analyzed individually without assuming a consistent SD.

among different wells of the culture plate, we determined the fold increase in luciferase activity at each time point relative to the luciferase activity at 24 h after transfection. We also used NS5-C709A mutants with inactivated RNA polymerase [20] as negative controls.

The averages of the three independent experiments showed that the fold increase in luciferase activity was approximately 2.1-fold higher with the Cosmopolitan virus (Gluc-CCCC) than with the Asian-I virus (Gluc-AAAA; P < 0.0001 at 120 h after transfection; Fig 7B, left panel). When the coding region of NS1, NS2A, and NS2B of the Asian-I type was replaced with that of the Cosmopolitan type (Gluc-CAAA), the fold increase in luciferase activity was higher than that with Gluc-AAAA (P = 0.0222, Fig 7B, left panel). Replacement of the coding region of NS1, NS2A, and NS2B of the Cosmopolitan type with that of the Asian-I type reduced the fold increase in luciferase activity by approximately 26.8% (Gluc-ACCC) when compared to the Cosmopolitan type (Gluc-CCCC; P = 0.0153 at 120 h after transfection).

On the other hand, all NS5-C709A mutants (Fig 7B, right panel) showed decreased fold increases in luciferase activity when compared to the wild-type viruses (Fig 7B, left panel) at 72 h post-transfection. At 120 h after transfection, there were statistically significant differences between the wild-type viruses and the NS5 mutants of Gluc-CAAA and Gluc-CCCC (Gluc-CAAA vs. Gluc-CAAA (C709A), P = 0.0056; Gluc-CCCC vs. Gluc-CCCC (C709A), P = 0.0029). Although the differences in Gluc-AAAA and Gluc-ACCC did not reach statistical significance (Gluc-AAAA vs. Gluc-AAAA (C709A), P = 0.384; Gluc-ACCC vs. Gluc-ACCC (C709A), P = 0.1106), the fold increases were always higher in the wild type viruses than in the NS5 mutants at 120h after transfection in each of the three independent experiments and showed statistically significant differences in Gluc-AAAA in the Experiment 1 (Gluc-AAAA vs. Gluc-AAAA (C709A), P = 0.02159) and in Gluc-ACCC in the Experiment 1 (Gluc-ACCC vs. Gluc-ACCC (C709A) and the Experiment 2 (Gluc-ACCC vs. Gluc-ACCC (C709A), P<0.0001) (S3 Fig). It should be noted that the trends in the differences in luciferase activity were the same among the C709A mutant viruses as the wild-type viruses (Gluc-CCCC (C709A) vs. Gluc-AAAA (C709A), P = 0.0199 at 120 h after transfection), although the differences among the mutant viruses were smaller than those among the wild-type viruses (Figs 7B and S3). Although it was confirmed that the C709A mutant knocked out virus propagation capability (Fig 2B and 2C), it is possible that the mutant polymerase still possessed leaky activity. Taken together, these results indicate that both the NS1-NS2B and NS3-NS5 regions are important for efficient replication of the virus.

### *In vivo* virus growth kinetics in IFNR-KO mice

The *in vivo* growth kinetics of viruses generated by the CPER method were analyzed using IFNR-KO mice. First, parental Asian-I and Cosmopolitan viruses were diluted to the concentrations of $1.0 \times 10^6$, $1.0 \times 10^5$, and $1.0 \times 10^4$ copies/mL, then 500 μL of each virus solution was injected intraperitoneally into the mice. Three mice were used for each virus dose. The body weight of the animals was measured daily for 14 days (S4 Fig). In addition, blood samples were taken on days 1, 4, 6, 10, and 13 after inoculation, and the amounts of viral RNA in the serum were measured. The amount of viral RNA in the blood of Asian-I virus (AAAAA)-infected animals reached a peak at the $4^{th}$ or $6^{th}$ day after inoculation, then subsequently decreased (Fig 8A). All but one of the mice infected with the Asian-I virus at a dose of $5.0 \times 10^4$ copies/mouse recovered (Fig 8B). In contrast, the amount of viral RNA in the blood of Cosmopolitan virus (CCCCC)-infected animals reached a peak on the $6^{th}$ day after inoculation, and all of the mice succumbed to infection without showing a decrease in the viral RNA levels in the blood (Fig 8A and 8B). Since the same trend was observed for all three inoculation doses, we decided to conduct subsequent experiments with an inoculation dose of $5.0 \times 10^5$ copies/mouse.

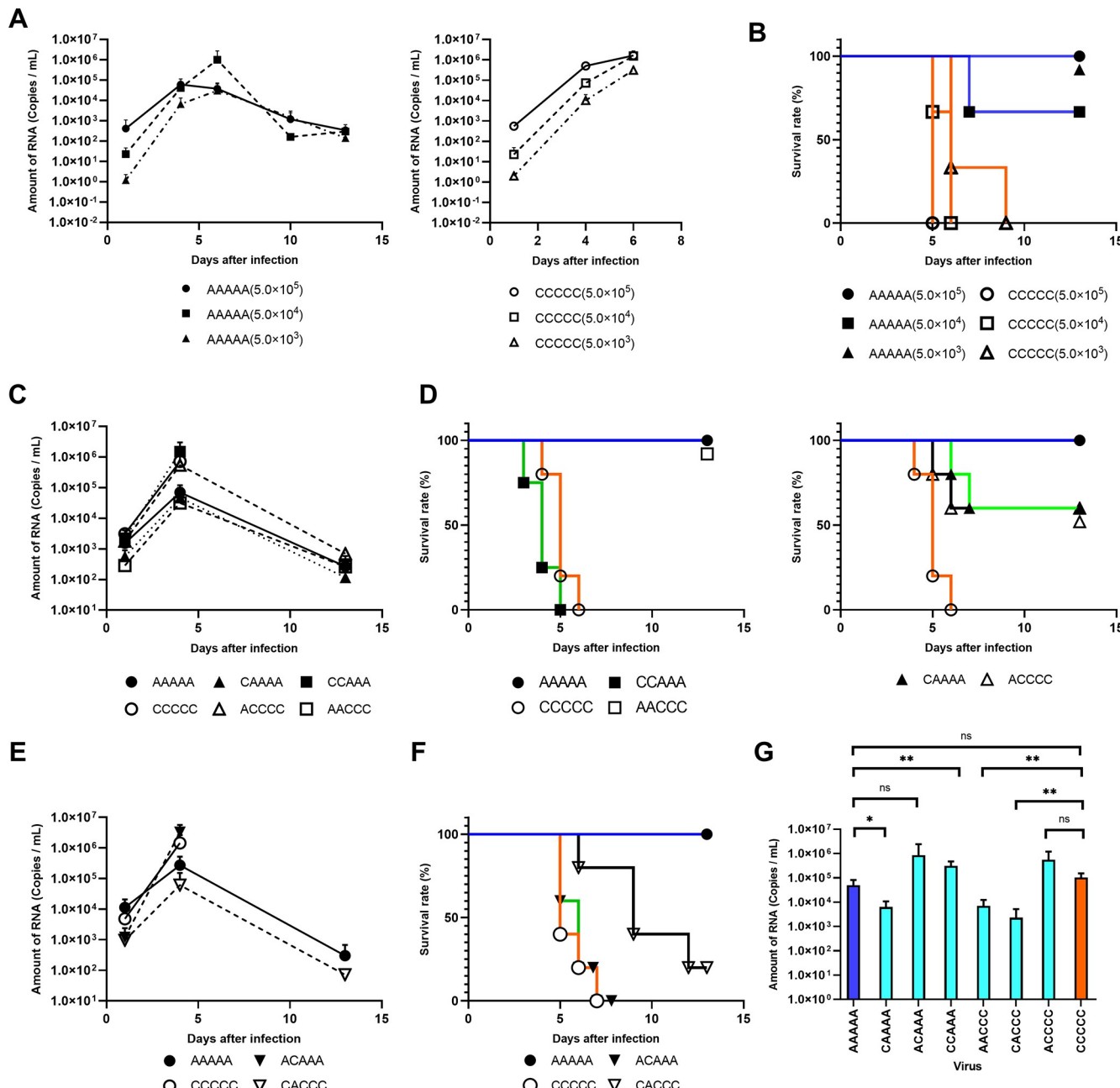

**Fig 8. Virulence of the recombinant viruses in the interferon receptor knock-out (IFRKO) mice.** (A) RNA levels in the blood were measured by real-time RT-PCR after inoculation of the IFNR-KO mice with DENV2 produced by the CPER method. On the graph on the left, results from days 1, 4, and 6 are shown as the means of triplicate samples, except for AAAAA ($5.0 \times 10^4$ copies/mouse) at days 10 and 13, the data of which are shown as the means of duplicate samples. In the graph on the right, data indicate the means of triplicate samples. (B) Survival curves of the mice shown in (A). By the log-rank test: P = 0.0003 (whole sample); P = 0.0253 ($5.0 \times 10^5$); P = 0.0295 ($5.0 \times 10^4$); and P = 0.0224 ($5.0 \times 10^3$). (C) Amount of RNA in the blood on days 1, 4, and 13, and the survival curves after inoculation with DENV2. The numbers of samples were as follows: AAAAA: n = 3 on day 1, and n = 5 on days 4 and 13; CCCCC: n = 3 on day 1, and n = 4 on day 4; CAAAA: n = 3 on days 1 and 13, and n = 5 on day 4; ACCCC: n = 3 on days 1 and 13, and n = 4 on day 4; CCAAA: n = 3 on day 1, and n = 4 on day 4; and AACCC: n = 3 on day 1, and n = 5 on days 4 and 13. (D) Survival curves of the mice shown in (C). The number of samples was 5 in each group, except for CCAAA, the data of which are shown as the means of quadruplicate samples. By the log-rank test: P = 0.002 (AAAAA vs. CCCCC, AAAAA vs. CCAAA, and CCCCC vs. AACCC); P = 0.1343 (AAAAA vs. CAAAA; not significant); and P = 0.0273 (CCCCC vs. ACCCC). (E, F) Blood RNA levels on days 1, 4, and 13, and the survival curves after DENV2 inoculation. The numbers of samples were as follows: AAAAA: n = 3 for all time points; CCCCC: n = 5 for all time points; ACAAA: n = 5 for all time points; and CACCC: n = 5 on days 1 and 4, and n = 2 on day 13. By the log-rank test: P = 0.0103 (AAAAA vs. CCCCC); P = 0.0120 (AAAAA vs. ACAAA); P = 0.0088 (CCCCC vs. CACCC); and P = 0.011 (ACAAA vs. CACCC). (G) Blood RNA levels on day 4 after DENV inoculation. The means and standard deviation of triplicate samples are shown. Results with statistically significant differences by the t test are indicated by asterisks (*: P < 0.05, **:P < 0.01).

Based on the results of the FFA (Fig 4) and the reporter assays (Figs 6 and 7), eight viruses, including the parental strains, were selected for the subsequent animal experiments. Five mice were used for each virus. As shown in Fig 8C, we found that the viruses with the Cosmopolitan structural proteins and NS1, NS2A, and NS2B (CCCCC and CCAAA) resulted in a higher amount of RNA in the blood and a higher lethality rate (Fig 8D, left panel) when compared to the viruses with the Asian-I structural proteins and NS1, NS2A, and NS2B (AAAAA and AACCC; Fig 8C, D, left panel). With the structural protein-swapped viruses, the amounts of RNA in the blood were almost the same as those of their parental viruses (Fig 8C; comparison between ACCCC with CCCCC, and CAAAA with AAAAA), but the lethality of these recombinant viruses was intermediate between those of the two parental viruses (Fig 8D, right panel). For the NS1, NS2A, and NS2B protein-swapped viruses, the amount of RNA in the blood of mice infected with the Asian-I virus with Cosmopolitan NS1, NS2A, and NS2B proteins (ACAAA) was as high as that in the mice infected with the parental Cosmopolitan (CCCCC) virus. Similarly, the Cosmopolitan virus with Asian-I NS1, NS2A, and NS2B proteins (CACCC) resulted in low levels of viral RNA in the blood, similar to the parental Asian-I (AAAAA) virus (Fig 8E). However, the lethality rates of these recombinant viruses were intermediate between those of the two parental viruses (Fig 8F). Finally, we simultaneously compared the viral loads of the eight viruses on the 4th day after inoculation, when the amount of RNA in the blood peaked. We confirmed that the amount of RNA was higher for viruses with the Cosmopolitan NS1, NS2A, and NS2B proteins (Fig 8G). These results indicated that NS1, NS2A, and NS2B proteins affect viral replication, and that swapping of the structural and NS1, NS2A, and NS2B proteins completely replaced the replication and pathogenicity of the viruses in mice.

## Structural analysis of the Asian-I and Cosmopolitan E proteins

The nucleotide sequences of the Th16-005DV2 (Asian-I) and Th16-026DV2 (Cosmopolitan) E protein-coding regions differ by 131 bases out of the total 1485 bases. In the amino acid sequences, 11 out of the 495 residues at positions 52, 71, 83, 141, 149, 226, 228, 346, 390, 462, and 484 in the E protein differ between the Asian-I and Cosmopolitan viruses. These mutations are also present in two additional Asian-I and two additional Cosmopolitan viruses (Fig 1). These mutations are located at positions 52, 141, and 149 in Domain-I, at positions 71, 83, 226, and 228 in Domain-II, at positions 346 and 390 in Domain-III, and at positions 462 and 484 in the transmembrane region (Fig 9A). In addition, homology modeling of the neighboring two sets of the E and M protein hetero-tetramer structure based on the DENV2 structure determined by cryo-electron microscopy (3J27 in the Protein Data Bank [22]) revealed that five mutations at positions 52, 71, 83, 226, and 228, one Domain-I and four Domain-II mutations, were located near or on the interface between the two E protein dimers (Fig 10A). Among these five mutations, those at 71 and 228 possessed negative charges in the Asian-I virus, but were neutral in the Cosmopolitan virus, while those at 83 and 226 possessed positive charges in the Asian-I virus, but were neutral in the Cosmopolitan virus. In addition, these five mutations are well conserved among the 894 Asian-I viruses reported from 1964 to 2020, and in the 662 Cosmopolitan viruses reported from 1974 to 2021 (S5 Fig and S2 Table). Homology modeling of the neighboring two sets of the E and M protein hetero-tetramer of the Asian-I virus further suggested that a hydrogen atom in an amino group of the lysine at position 83 of E protein in one E and M hetero-tetramer was positioned very closely to an oxygen atom in a carboxyl group of the glutamic acid at position 228 of E protein in another E and M hetero-tetramer (Fig 10B).

We therefore generated an additional chimeric virus carrying the aforementioned five amino acid mutations at positions 52, 71, 83, 226, and 228 located at the interface between the

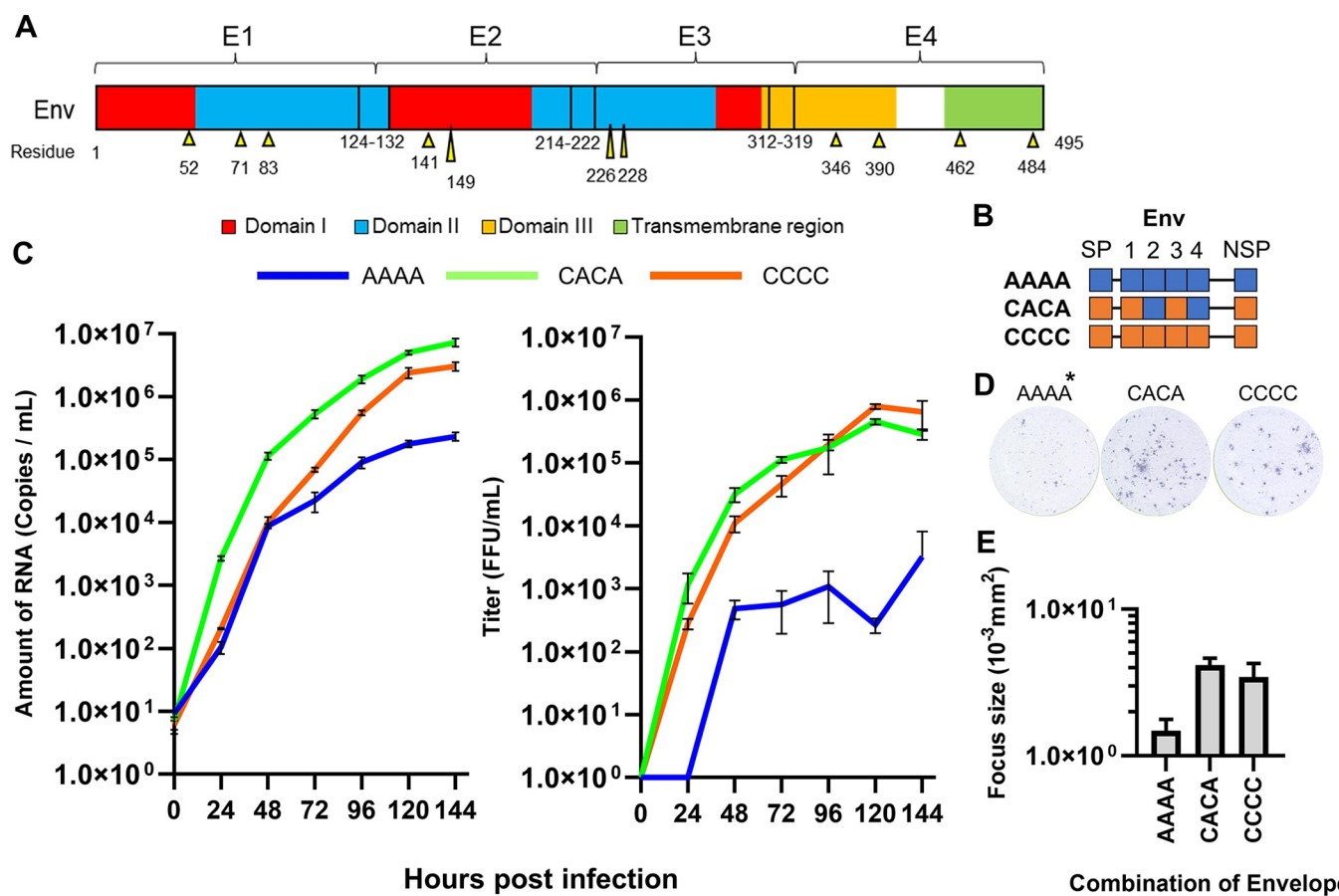

**Fig 9. Characteristics of viruses with recombinant envelope proteins.** (A) The mutation sites between Asian-I and Cosmopolitan viruses are shown. The divided envelope regions are denoted as E1, E2, E3, and E4. The domain-I region is red, the domain-II region is blue, the domain-III region is yellow, and the transmembrane region is yellow-green. The mutation sites are indicated by yellow arrows. (B) Schematic representation of viruses containing chimeric envelope proteins from two different parental strains. The fragments derived from the Asian-I [A] genotype and Cosmopolitan [C] genotype are shown in blue and orange, respectively. SP: capsid and prM; NSP: nonstructural proteins. (C) Growth curves of the recombinant viruses in Vero cells infected with virus samples at a multiplicity of infection of 0.5 copies/cell. Culture supernatants were harvested every 24 h, and assayed for the levels of viral RNA and the infectious titers. The means and standard deviation of triplicate samples are shown. (D) Representative images of the foci of chimeric viruses. *: AAAA, CACA, and CCCC indicate the chimeric envelope proteins. The structural and non-structural proteins of AAAA are derived from the Asian-I type, and those of CACA and CCCC are from the Cosmopolitan type. (E) Focus sizes of the chimeric viruses. The means and standard deviation of triplicate samples are shown.

E protein dimers. We divided the E protein into four parts, E1, E2, E3 and E4, and generated a virus with a chimeric E protein with Cosmopolitan amino acids at five positions, 52, 71, 83, 226 and 228, near or at the E protein dimer interface, and Asian-I amino acids at the remaining six positions (Fig 9A and 9B). Virus stocks were generated, and equal amounts of viruses were inoculated into Vero cells. As shown in Fig 9C, the amount of RNA and the infectious titer of the CC-CACA-CCCC virus were comparable to those of the parental Cosmopolitan virus. The focus size of this chimeric virus did not differ significantly from those of the parental strains (Fig 9D and 9E). Post-hoc sequence analysis of the virus stocks generated above confirmed the sequence authenticity of the viruses.

These results suggest that mutations in the five amino acids (at positions 52, 71, 83, 226, and 228) present in the E protein interface are, at least in part, responsible for the differences in replication, pathogenicity, and infectivity between the Asian-I and Cosmopolitan genotypes of DENV2.

## Discussion

In the present study, we found differences in the prM/E sequences between the Asian-I and Cosmopolitan types of DENV2 that determined virus infectivity, and differences in the sequences of NS1, NS2A, and NS2B proteins also affected virus replication in cell cultures and mice. Furthermore, we found that particular combinations of structural and NS1, NS2A, and NS2B proteins were responsible for the differences in viral pathogenicity in mice. Molecular modeling of the neighboring two sets of the E and M protein hetero-tetramer suggested that many of the mutations in the E protein responsible for the differences between Asian-I and Cosmopolitan viruses were located near or on the interface between the two E protein dimers (Fig 10A and 10B). It is thus possible that these mutations may affect conformational changes of E protein upon infection. Similarly, it was previously reported that the Cosmopolitan type TW2015 strain carrying the same mutations as Th16-026DV2 also showed high pathogenicity in interferon I/II receptor KO mice [23].

The Asian-I genotype of DENV2 has long been dominant in Thailand, but we recently reported the emergence of Cosmopolitan genotypes of DENV2 in 2016 in Thailand [10]. Since then, the Cosmopolitan genotype has been co-circulating with the Asian-I genotype in

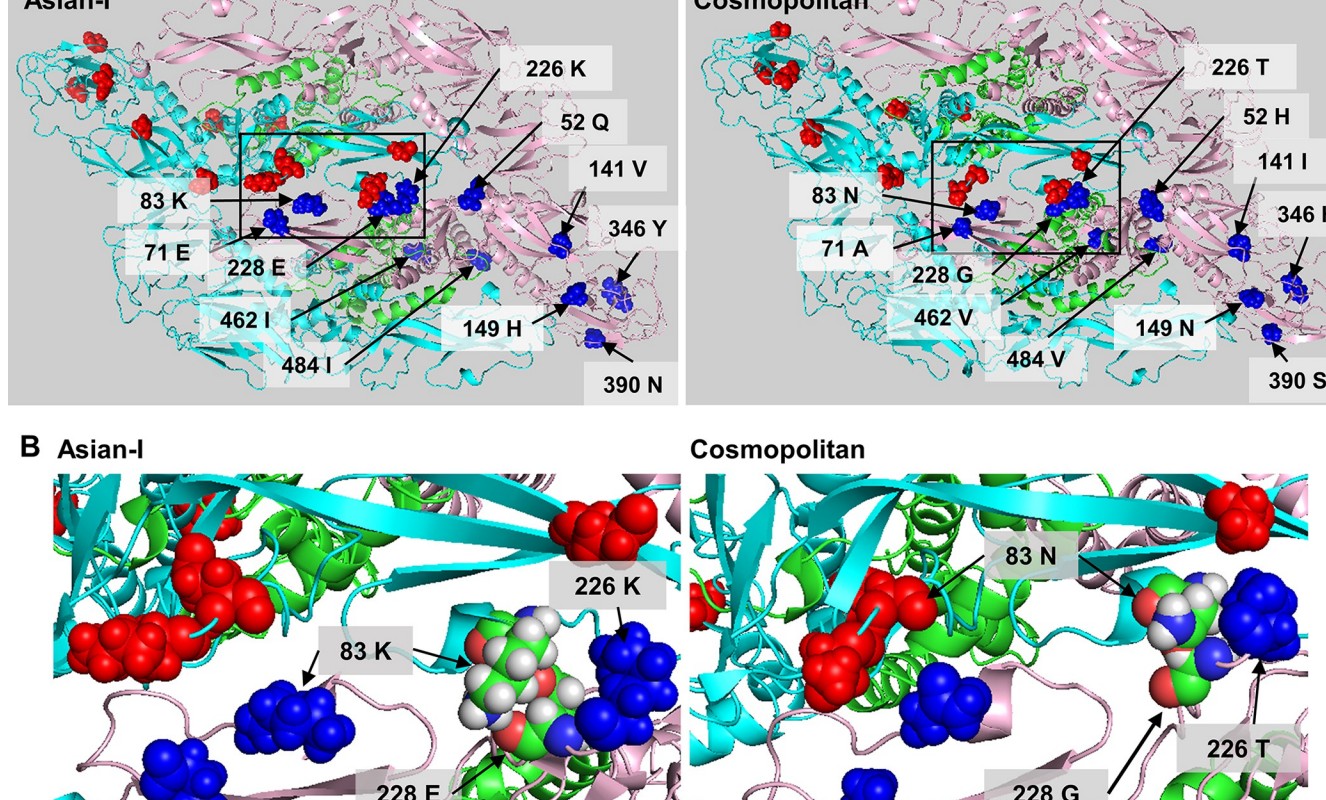

**Fig 10. Molecular modeling of neighboring two sets of the E and M protein hetero-tetramer of Asian-I and Cosmopolitan viruses.** (A) The mutation sites in the envelope protein tetramers of the Asian-I and Cosmopolitan viruses are shown. Both the red and blue spheres indicate mutation sites in the envelope protein. Cyan and pink regions are the envelope protein, and the green region is the M protein. (B) Enlarged images of the black-framed areas in A. Amino acid residues at positions 83 and 228 are colored as follows: green indicates carbon atoms; gray indicates hydrogen atoms; blue indicates oxygen atoms; and red indicates nitrogen atoms. These figures were generated using PDB ID: 3J27 as a template for homology modeling.

Thailand [9,24]. The Cosmopolitan genotype has been reported to be circulating in many parts of the world [6–8,25]. There have been many cases of severe dengue fever caused by the Cosmopolitan virus [26,27]. In fact, Cosmopolitan Th17-077DV2 [10] has been isolated from a patient with shock symptoms. Genotypic shifts have been attributed to the fact that even strains of the same serotype have different properties in terms of infectivity, pathogenicity, and antigenicity, some of which can lead to increased viral fitness. Viral fitness is determined by a variety of factors, including the genetics of the virus strain, its ability to propagate via vectors, and its interaction with various molecules in the human body [28]. In the case of DENV1, lineage 6 of genotype V has been found in São José do Rio Preto, Brazil. Interestingly, when the faster-growing lineage 1 virus appeared [29], it did not replace the lineage 6 virus; rather, it eventually disappeared, and it was speculated that despite its fast growth, the lineage 1 virus stimulated the immune response so strongly that it was eradicated by herd immunity. In contrast, the Asian-I and Cosmopolitan types are thought to have achieved genotypic appropriateness in Thailand, and various factors are expected to influence future epidemics caused by these viruses in the future.

In the present study, we used the CPER method to generate recombinant viruses (Fig 4). The CPER method is versatile and enabled the quick generation of numerous plasmids carrying recombinant virus genomes. However, post-hoc sequence analysis of the recovered virus stocks revealed the presence of an unexpected amino acid mutation at position 202 in the E protein of the CCACC virus even though the plasmids used for this CPER reaction did not carry this mutation. It is likely that the mutation occurred during the 20 cycles of the PCR reaction in the CPER method. Fortunately, the exclusion of this particular virus did not affect the conclusions of our study. However, the error-prone nature of the CPER method is a limitation of the method, and it is important to be aware of the possibility of unintended mutations.

The FFA results revealed differences in the focus sizes and infectious titers among the chimeric recombinant viruses with swapped structural proteins and/or non-structural proteins NS1, NS2A, and NS2B (Fig 4). When the prM and E proteins were derived from different genotypes, the infectious titers, and sometimes the focus sizes, decreased when compared to those of the parental Asian-I genotype, which showed lower titers and formed smaller foci than the Cosmopolitan genotype (Fig 5). Furthermore, SRIPs with chimeric prM/E showed no luciferase activity (Fig 6), indicating a loss of infectivity. These results suggest that the combination of prM/E sequences has a very important role in the infectivity of the virus. These results are in good agreement with those of a previous report showing the high infectivity of a chimeric virus containing both prM and E proteins of Cosmopolitan DENV2 [30].

The nucleotide sequences of the Th16-005DV2 (Asian-I) and Th16-026DV2 (Cosmopolitan) structural protein regions differed by 41 out of the 498 bases in the prM. In the amino acid sequences, 7 out of 166 residues in the prM differed. These substitutions were also found in the two additional Asian-I and two additional Cosmopolitan viruses shown in Fig 1. The prM protein mutations were located at positions 15, 16, 29, 52, 82, 127, and 148. Among these seven positions, position 52 was always occupied by a lysine in the 894 Asian-I viruses reported from 1964 to 2020, and by an asparagine in the 662 Cosmopolitan viruses reported from 1974 to 2021 (S5 Fig). It has been established that prM and E are closely involved in virion maturation, which requires a major rearrangement of M and E proteins during the process of immature-to-mature particle transformation [31]. The prM protein is cleaved by furin in the *trans*-Golgi network, and the proteolytic product, pr, remains associated with E to prevent membrane fusion until the progeny virion is released to the extracellular milieu [32]. The 3D structure of the pr-E complex [31] revealed that the mutation position 52 in pr exists in close vicinity of the mutation position 71 of E. Homology modeling of the pr-E complex of the Asian-I type suggested that a hydrogen atom in an amino group of the lysine at position 52 of

pr was positioned very closely to an oxygen atom in a carboxyl group of the glutamic acid at position 71 of E (Fig 11A and 11B). In contrast, there was an apparent space between the asparagine at position 52 of pr and the alanine at position 71 of E in the Cosmopolitan type (Fig 11A and 11B). Taken together with the findings for certain amino acids shown in Figs 5 and 6, these results indicated the possibility that the protein-protein interactions between pr and E are significant in maintaining proper infection processes. It should also be noted here that among the mutation positions in E, positions 71, 141, and 390 have been reported to be important for evading neutralizing antibody responses, and they may be sites under immune pressure [30].

The nucleotide sequences of the Th16-005DV2 and Th16-026DV2 non-structural protein genes differed by 101 out of the 1056 bases in NS1. In the amino acid sequences, differences were observed at 14 out of 352 residues in NS1. The mutations of NS1 were located at positions 50, 80, 117, 128, 129, 131, 174, 177, 222, 247, 264, 272, 281, and 286. The mutations at positions 50, 80, 117, 128, 129, and 131 were located in the α/β subdomain, those at positions 174 and 177 were located in the connector subdomain, and those at positions 222, 247, 264, 272, 281, and 286 were located in the β-ladder domain (Fig 12A). Various studies have been conducted on nonstructural proteins, particularly the NS1 protein, which is found in virus-induced vesicle compartments that house early viral RNA replication and viral replication complexes in the cell [33]. Also, the levels of NS1 proteins circulating in the blood during the acute phase of infection have been reported to increase in correlation with peak viremia and disease severity [34]. Although further studies are necessary, the *in vivo* differences might have been due, at least partly, to the significant impact of NS1 proteins in increasing the pathogenicity. Also, the lysine residues at positions 272 and 324 have been reported to contribute to the instability of the NS1 glycoprotein in human hepatocyte HuH7 cells [35]. A lysine residue was almost always found at position 272 in the Asian-I viruses (S5 Fig), while arginine was found at position 272 in the Cosmopolitan virus used in this experiment. It is thus possible that the altered properties of NS1 may have played a role in the altered *in vivo* disease course. It should also be mentioned that a previous co-immunoprecipitation study has reported interactions between the NS1 and E proteins [36]. Since it has been reported that the NS1 protein resides on the same side of the endoplasmic reticulum lumen as the E protein before it is released [37,38], it is possible that interactions between the NS1 and E proteins might be involved in the combined effects of fragments 1 and 2 on the increased pathogenesis of the Cosmopolitan virus.

NS2A is known to have important roles in both viral RNA synthesis and virion assembly, and to antagonize host innate immune responses [39,40]. The nucleotide sequences of the Th16-005DV2 and Th16-026DV2 non-structural protein genes differed by 76 out of the 654 bases in NS2A. In the amino acid sequences, differences were observed at 8 out of 218 residues in NS2A. These NS2A mutations were located at positions 5, 51, 57, 65, 120, 158, 171, and 215 (Fig 12B). The mutations located at positions 158 and 171 were in the transmembrane segments, that at position 215 was located in the C-terminal cytoplasmic domain, and those at positions 5, 51, 57, 65, and 120 were located in the endoplasmic reticulum lumen [41]. Therefore, the mutations appear to be concentrated on the endoplasmic reticulum lumen side, which may affect NS2A interactions with certain host factors.

NS2B is a transmembrane protein that functions as a cofactor for viral NS3 protease [42]. The nucleotide sequences of the Th16-005DV2 and Th16-026DV2 non-structural protein genes differed by 38 out of the 390 bases in NS2B. In the amino acid sequences, differences were observed at 7 out of 130 residues in NS2B. These NS2B mutations were located at positions 21, 57, 59, 60, 63, 65, and 112 (Fig 12C). The mutations located at positions 21 and 112 were in the transmembrane domain, and those at positions 57, 59, 60, 63, and 65 were in the cytoplasmic region. Structural analysis of the NS2B-NS3 complex revealed that the mutations

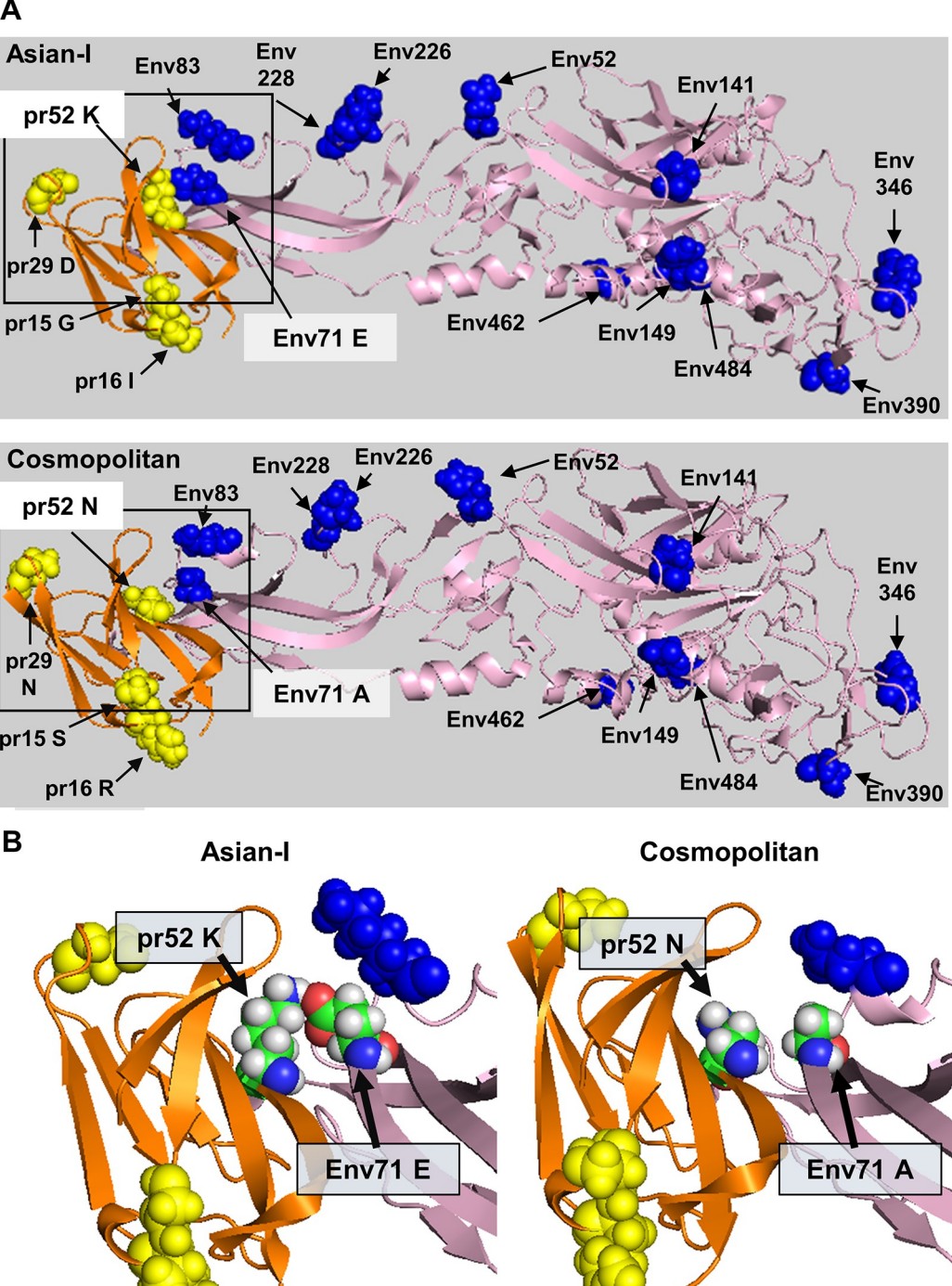

**Fig 11. Molecular modeling of the Asian-I and Cosmopolitan envelope and pr complex.** (A) The mutation sites in the pr and envelope proteins of the Asian-I and Cosmopolitan viruses are shown. The orange region is the pr protein, and the pink region is the envelope protein. The yellow spheres indicate mutation sites in the pr protein. The blue spheres indicate mutation sites in the envelope protein. (B) The mutation sites of position 52 of pr and position 71 of envelope are shown. Amino acid residues at positions 52 of pr and 71 of envelope are colored as follows: green indicates carbon atoms; gray indicates hydrogen atoms; blue indicates oxygen atoms; and red indicates nitrogen atoms. These figures were generated using PDB ID: 3J27 and 3C5X as templates for homology modeling.

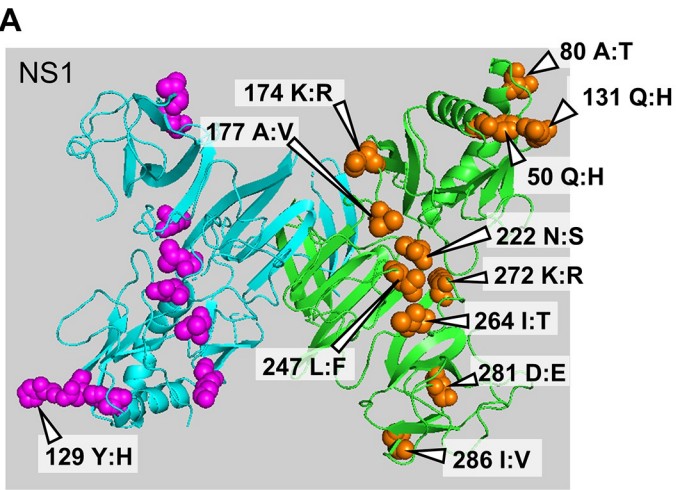

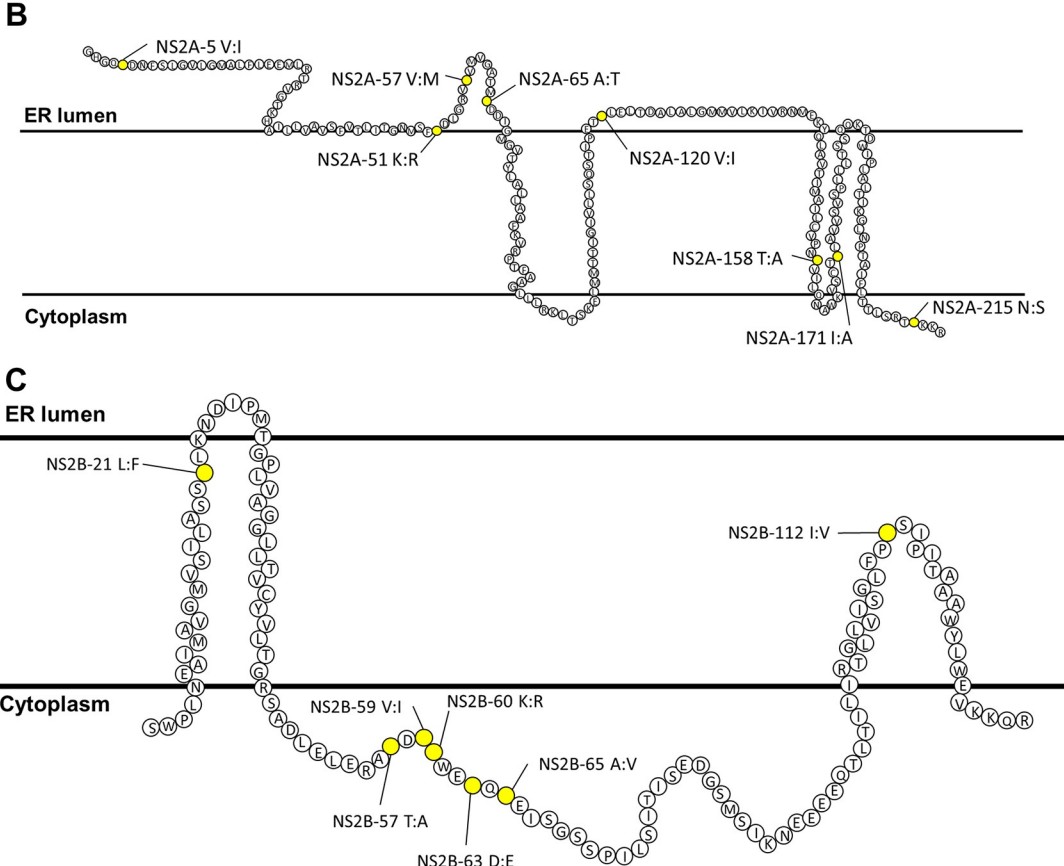

**Fig 12. Mutation sites in the non-structural proteins.** (A) The mutation sites in the Asian-I and Cosmopolitan viruses are shown on the 3D structure of the NS1 protein dimer (PDB ID: 4O6B). Both the orange and purple spheres indicate mutation sites in the NS1 protein. A region spanning amino acid positions 108 to 128, including mutation cites at positions 117 and 128, is not shown due to the lack of structural information in the template. (B, C) A topology model of the DENV2 NS2A and NS2B proteins on the endoplasmic reticulum membrane. The mutation sites in the NS2A and NS2B proteins (yellow points) of the Asian-I and Cosmopolitan viruses are shown. These topology models were drawn based on previous articles referred to in the text (reference numbers 41 to 43).

at positions 57–65 were located on the loop structure of the cytoplasm side; this loop is also the binding site for NS3, and is thought to play a pedestal-like role [43,44].

The results obtained in the present study showed that the viral gene expression level was significantly changed by the recombination of the non-structural proteins NS1, NS2A, and NS2B (Fig 4B), but that these proteins were still insufficient to completely switch the phenotypes of the parental strains (Fig 7B). This suggests that the viral gene expression level is determined by the interaction of these non-structural proteins with structural proteins or other non-structural proteins. Since this study was conducted only on DENV2, it is unknown whether similar results would be obtained for other viruses. Analysis of gene function for the other serotypes is required as a topic of future research.

In addition, the results of the animal experiment showed that the recombination of NS1, NS2A, and NS2B with the Cosmopolitan virus resulted in increased RNA levels and increased lethality, and had a stronger effect on phenotypic changes than that seen in the *in vitro* experiments. However, these results were observed in mice, and they do not necessarily indicate that similar symptoms would be seen in humans. To evaluate the immune responses to the Asian-I and Cosmopolitan genotypes of DENV2 in humans, further experiments that reproduce human immune mechanisms are necessary. Understanding of the interactions among these proteins, and the changes in their properties caused by amino acid mutations may help us to elucidate the pathogenesis of DENV and factors that contribute to its severity.

## Supporting information

**S1 Table. Primer list.**
(PDF)

**S2 Table. Accession number list.**
(PDF)

**S1 Fig. Levels of viral RNA in the cell supernatants.** (A) Levels of viral RNA in the cell supernatants after the transfection of CPER products. (B) Levels of viral RNA in the cell supernatants after viral infection with the supernatants of the transfected cells.
(PDF)

**S2 Fig. Characteristics of the parental and chimeric DENV2 between the Cosmopolitan and Asian-I genotypes generated from an independent CPER reaction.** (A) Infectious titers of an independent lot of stock viruses. (B) Focus-forming unit of an independent lot of stock viruses 3 days after the infection of Vero cells at a multiplicity of infection of 0.5 copies/cell. The number of seeding Vero cells was $4.0 \times 10^4$ cells/well. These experiments were performed using the method described in the "Focus-forming assay" section of the Materials and Methods.
(PDF)

**S3 Fig. Replicon assay.** (A, B, C) Results of three independent experiments on Replicon assay. The fold increases in luciferase activity when compared to the activity at 24 h after the transfection of DENV2 replicons are shown as the means and standard deviation (SD) of triplicate samples. Statistical significance of the differences were tested using the Holm-Sidak method with alpha = 0.05. Each row was analyzed individually without assuming a consistent SD.
(PDF)

**S4 Fig. Weight changes in the mice during the experimental infection.** (A) Body weight changes of the mice shown in Fig 8A and 8B. (B) Body weight changes of the mice shown in Fig 8C and 8D. (C) Body weight changes of the mice shown in Fig 8E and 8F. In all data, "-"

indicates after euthanization. Parts with no data are left blank.
(PDF)

**S5 Fig. Differences in amino acid residues between the Asian-I and Cosmopolitan types.**
Differences in amino acid residues in two structural proteins and three non-structural proteins
between the Asian-I and Cosmopolitan types. Amino acid residues found in Th16-005DV2
and Th16-026DV2 are shown in bold font. For each position, the amino acid residue and its
frequency (percentage) in the sequences are shown. The number of sequences used was 894
for the Asian-I type, and 662 for the Cosmopolitan type (S2 Table).
(PDF)

## Acknowledgments

We thank Kumi Yamamoto for her help.

## Author Contributions

**Conceptualization:** Akatsuki Saito, Emi E. Nakayama, Tatsuo Shioda.

**Data curation:** Yoshihiro Samune, Akatsuki Saito, Ritsuko Koketsu, Osamu Kotani, Emi E. Nakayama, Tatsuo Shioda.

**Formal analysis:** Yoshihiro Samune, Juthamas Phadungsombat, Masaru Yokoyama, Emi E. Nakayama, Tatsuo Shioda.

**Funding acquisition:** Yoshihiro Samune, Emi E. Nakayama.

**Investigation:** Yoshihiro Samune, Narinee Srimark, Juthamas Phadungsombat, Masaru Yokoyama, Osamu Kotani, Atsushi Yamanaka.

**Methodology:** Akatsuki Saito, Tadahiro Sasaki, Ritsuko Koketsu, Masaru Yokoyama, Atsushi Yamanaka, Takeshi Kurosu.

**Project administration:** Hironori Sato, Tatsuo Shioda.

**Resources:** Narinee Srimark, Atsushi Yamanaka, Saori Haga, Toru Okamoto, Takeshi Kurosu.

**Supervision:** Akatsuki Saito, Tadahiro Sasaki, Ritsuko Koketsu, Hironori Sato, Toru Okamoto, Emi E. Nakayama, Tatsuo Shioda.

**Validation:** Akatsuki Saito, Tadahiro Sasaki, Masaru Yokoyama, Osamu Kotani, Emi E. Nakayama, Tatsuo Shioda.

**Visualization:** Yoshihiro Samune, Tadahiro Sasaki.

**Writing – original draft:** Yoshihiro Samune, Masaru Yokoyama.

**Writing – review & editing:** Akatsuki Saito, Juthamas Phadungsombat, Hironori Sato, Atsushi Yamanaka, Toru Okamoto, Takeshi Kurosu, Emi E. Nakayama, Tatsuo Shioda.

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
