## [Decision Letter · Decision Letter 0]

6 Aug 2023

Dear Dr. Shioda,

Thank you very much for submitting your manuscript "Genetic regions affecting the replication and pathogenicity of dengue virus type 2" for consideration at PLOS Neglected Tropical Diseases. As with all papers reviewed by the journal, your manuscript was reviewed by members of the editorial board and by several independent reviewers. In light of the reviews (below this email), we would like to invite the resubmission of a significantly-revised version that takes into account all of the reviewers' comments. 

The reviewers raised multiple relevant points regarding your study that must be addressed for publication. First, all reviewers noted you did not include figure legends which is a major issue when reviewing the manuscript. Second, all reviewers questioned why virus stocks were not generated, sequenced, and used to compare viral growth. It will be critical to complete these experiments. Finally, reviewer 1 makes a good point regarding proper controls for all experiments. Given these major criticisms and those below, I recommend you address all of the reviewers points for resubmission.

We cannot make any decision about publication until we have seen the revised manuscript and your response to the reviewers' comments. Your revised manuscript is also likely to be sent to reviewers for further evaluation.

Sincerely,

Kenneth A. Stapleford, Ph.D.

Guest Editor

Abdallah Samy

Section Editor

Reviewer's Responses to Questions

**Key Review Criteria Required for Acceptance?**

**Methods**

-Are the objectives of the study clearly articulated with a clear testable hypothesis stated?

-Is the study design appropriate to address the stated objectives?

-Is the population clearly described and appropriate for the hypothesis being tested?

-Is the sample size sufficient to ensure adequate power to address the hypothesis being tested?

-Were correct statistical analysis used to support conclusions?

-Are there concerns about ethical or regulatory requirements being met?

Reviewer #1: The study aims at mapping the genetic determinants responsible for the differences in cell culture growth of the Cosmopolitan and Asian I genotypes of Dengue virus 2 (DENV2). To this end it follows a genetic approach based on the construction of chimeric viruses and the assessment of their phenotypes in vitro and in vivo. The genetic system relies on the production of virus RNA genomes from a CMV promoter in transfected cells. While the system appears versatile to generate the large number of chimeras used in the study, conclusions made from the generation virus genomes based on RNA transcription in the cell nucleus need to consider the limitations of the system.

(1) How do the authors control for differences in transfection efficiencies? Throughout the manuscript, the comparisons of virus growth in culture should be assessed using virus stocks to perform growth curves as shown in Figure 1. This is specially necessary to compare Cosmopolitan and Asian I recombinant types in Figure 3, and combination of envelope fragments in Figure 9.

(2) Figure 2 and lines 226-227. Please describe what the negative control consists of. Mock DNA transfections may not be appropriate controls. Instead, a control construct bearing a a mutation in the NS5 RdRp catalytic site would be more meaningful.

(3) Figure 3 and lines 233-235. The rationale of the approach is not properly described in the text. In panel A, are foci size measured in cells infected with the supernatants of cells transfected with CPER products? If this is the case, were cells infected at the same MOI? In panel B, while quantification of RNA copies in the supernatant shows differences within one order of magnitude for the Cosmopolitan type compared to the Asian I type (left), differences in infectious virus production are at least >20-fold (right). This is also an indication of the impact of the genetic background on virus infectivity that is overlooked in the text. Finally, and referred to comment (1), curves of RNA or virus production as a function of time after transfection of CPER products are not appropriate to measure growth kinetics.

(4) Figure 7. In the replicon assay using Gaussia luciferase as a reporter, lines 291-294 claim that luciferase levels measured in the supernatant of transfected cells as of 48 hours post-transfection reflect authentic genome replication. Characterization of the assay previously reported by the authors (DOI: 10.7883/yoken.67.209) showed that differences in luciferase levels for the wild type construct and an NS5 mutant control are < 10-fold. Therefore, it is arguable whether differences observed for the Cosmopolitan and Asian I types in the figure reflect functionality of the swapped NS1-2B fragment. The NS5 mutant construct needs to be included in the analysis. 

(5) Figure 8D and 8F. Lethality of the Asian I type is disparate between panels. While no lethality is registered in panel D, a 40% lethality is reported in panel F for the same virus. Since chimeric viruses result in intermediate lethality phenotypes, the differences between experiments for the same virus argue against the significance of conclusions.

Reviewer #2: Yes the study clearly articulates its hypothesis and is designed and executed to evaluate it.

Reviewer #3: (No Response)

**Results**

-Does the analysis presented match the analysis plan?

-Are the results clearly and completely presented?

-Are the figures (Tables, Images) of sufficient quality for clarity?

Reviewer #1: Figure legends are not provided in the current version of the manuscript thus information regarding statistical analyses (number of replicate experiments, tests applied, significance of the observed differences) is not available. It is therefore not possible to judge the soundness of the data.

Reviewer #2: Figures are good. I missed the figure legends.

Reviewer #3: (No Response)

**Conclusions**

-Are the conclusions supported by the data presented?

-Are the limitations of analysis clearly described?

-Do the authors discuss how these data can be helpful to advance our understanding of the topic under study?

-Is public health relevance addressed?

Reviewer #1: The limitations of the CPER method need to be clearly stated in the text as already commented above.

In addition, construction of homology models can serve as an approach to define likely interactions between residues. In these models whether two residues are interacting is based only in geometric criteria, i.e. the distance between residues in the model. In particular, for hydrogen bonding also the angles confine the possibility of interaction between residues. Therefore, assumptions made on the impact of residue differences between Cosmopolitan and Asian I types on envelope proteins maturation and dynamics (lines 417-424) should be toned down. Alternatively, molecular dynamics calculations can be performed to further support the proposed model.

Reviewer #2: Yes, the conclusions are for the most part supported by the data. See my specific points to see where they authors stray from that point.

Reviewer #3: (No Response)

**Editorial and Data Presentation Modifications?**

Reviewer #1: References are not uniformly formatted

Reviewer #2: (No Response)

Reviewer #3: (No Response)

**Summary and General Comments**

Reviewer #1: The manuscript by Samune and colleagues presents a systematic analysis of the viral determinants that account for the differences in the in vitro and in vivo fitness of the Cosmopolitan and Asian I genotypes of dengue virus serotype 2 (DENV2). The two genotypes of the virus co-circulate in Thailand together with viruses of the other three serotypes. The initial observation is that isolates of these two genotypes display differences in foci size in mammalian cells and growth kinetics in mosquito and mammalian cells, with the Cosmopolitan type showing bigger foci size and faster growth. Using a circular extension polymerase reaction (CPER) to construct a series of chimeric cDNAs of Cosmopolitan and Asian I types that launch full length RNA virus genomes from the CMV promoter the authors are able to narrow down the regions responsible for the observed differences to the structural proteins (C-prM-E) and non structural proteins (NS) 1, 2A, and 2B. Further dissection of the structural proteins coding region indicates that chimeric viruses bearing Cosmopolitan type prM and E proteins in the Asian I type background yield higher virus titers and form bigger foci compared to the parental Asian I type. The data also point to a necessary association between prM and E proteins that is also evidenced using a reporter replicon trans packaging system: replicon RNA is packaged into particles consisting of chimeras of the prM and E proteins, but reporter gene activity is only detected in cells infected with the assembled particles when particles carry prM and E of the homologous type. In turn, the CPER approach is also used to construct cDNA replicons carrying Gaussia luciferase in place of virus structural proteins to address the contribution of NS1, NS2A, and NS2B to the phenotype of Cosmopolitan and Asian I types. Results indicate that insertion of Asian I typr NS1-2B fragment in the Cosmopolitan type background results in impaired luciferase activity compared to the parental construct, while insertion of the Cosmopolitan type fragment in the Asian I type background enhances luciferase activity. In experimental infections of immunodeficient mice, the Cosmopolitan type virus reaches higher titers than the Asian I type and results in increased lethality. In this model, chimeric viruses bearing heterologous structural or NS1-2B fragments display phenotypes between the parental Asian I and Cosmopolitan types. Finally, the determinants responsible for the differences in the cell culture phenotype are pinpointed to five residues in the E coding sequence that differ between the two types. Mapping of residues into molecular models suggests an involvement of these residues in the interaction between E dimers on the virion surface and in the contact between E and prM.

Overall, the manuscript provides a descriptive analysis of virus phenotypes that follows a systematic characterization of chimeric constructs to identify a set of residues located in the structural proteins of the virus, namely a residue in prM and five residues in E, that account for the differences in growth kinetics, infectivity, and outcome of infection in animal models of the Cosmopolitan and Asian I types of DENV2. Insights into the mechanisms are only speculative and based on the construction of molecular homology models of the implicated proteins. Major concerns arise regarding the strength of data as the manuscript is missing figure legends describing the number of replicate experiments, data analysis, and the statistical tests applied.

Reviewer #2: In regions of southeast Asia, for example Thailand, there are circulating strains of all four dengue virus serotypes that expand and contract in the population. The authors of this manuscript examine the properties of two contemporary strains dengue 2 virus, the Asian-I genotype and the Cosmopolitan genotype, that circulate in Thailand. They demonstrate the superior growth in cell culture and in a mouse model of the Cosmopolitan strain. They go on to construct a series of chimeric Cosmopolitan – Asian viruses and map the regions and eventually some of the residues responsible for the enhanced replication phenotype. The authors conclude that prM-E need to be matched. They use the known structures to map the responsible residues and suggest that the Cosmopolitan strain may be less constrained in E-E interactions and this may result in enhanced replication. They also show that there are determinants in the nonstructural proteins NS1-NS2B from the Cosmopolitan strain also influence the enhanced replication. Collectively the manuscript is well written and the experiments are performed well.

Specific comments:

I did not find figure legends.

Lines 223-234: It would be best to directly compare the parental viruses with the cDNA-derived viruses to validate this statement.

Line 275: Define SRIP as not everyone will know this.

Line 358: A hydrogen bond would probably not contribute much to the stability of the E – E interactions.

Figure 5: In panel D the ACC virus has a larger focus size than CCC and yet CCC grows to higher titer (panel B). Can the authors address this?

Line 370: This line is an overstatement to say that these 5 amino acids are responsible for replication of dengue virus. It needs to be restated to be more accurate.

Line 385: Given the results presented by the authors, why doesn’t the Cosmopolitan strain outcompete the Asian-I strain? They should discuss this.

Line 441: Is there a clear connection between the strain of E and NS1 affecting replication? Perhaps this point should be discussed further if the authors can use their data to address this.

Reviewer #3: The manuscript by Samune et al. investigates the differences in replication and pathogenesis of DENV2 strains isolated in Thailand in 2016-2017. They found that the NS1-NS2A-NS2B genes dictated pathogenesis in mice.

Major comments:

1. Were the rescued virus stocks sequenced? This is critical.

2. Figure legends are missing. It was very hard to follow Fig. 4 without figure legends. What are the colors indicating? There are chimeras with A fragment of NS1-NS2A-NS2B with very high titers. Why would you want to highlight viruses with titers LOWER than A or HIGHER than C? This reflects viral incompatibilities. I would think you would want to highlight viruses with titers HIGHER than A or LOWER than C.

3. Which UTRs were used in the chimeras?

4. Focus forming assays on serum samples would be preferable to RNA levels.

5. In order to conclude that the 5 a.a. in the E protein interface are responsible for pathogenicity, mouse studies need to be performed. Alternatively, the language could be modified.

PLOS authors have the option to publish the peer review history of their article (what does this mean?). If published, this will include your full peer review and any attached files.

Reviewer #1: No

Reviewer #2: No

Reviewer #3: No
---

## [Decision Letter · Decision Letter 1]

9 Nov 2023

Dear Dr. Shioda,

Thank you very much for submitting your manuscript "Genetic regions affecting the replication and pathogenicity of dengue virus type 2" for consideration at PLOS Neglected Tropical Diseases. As with all papers reviewed by the journal, your manuscript was reviewed by members of the editorial board and by several independent reviewers. The reviewers appreciated the attention to an important topic. Based on the reviews, we are likely to accept this manuscript for publication, providing that you modify the manuscript according to the review recommendations. 

As you can see below, Reviewer 1 still has several points that should be addressed in the manuscript. Specifically, the point regarding technical vs. biological replicates is important to address. In addition, Figure 7B, which is currently a representative of several experiments, should be the average of all experiments if statistics will be shown. Given these issues, please update the figures accordingly and address the comments below.

Sincerely,

Kenneth A. Stapleford, Ph.D.

Guest Editor

Abdallah Samy

Section Editor

Reviewer's Responses to Questions

**Key Review Criteria Required for Acceptance?**

**Methods**

-Are the objectives of the study clearly articulated with a clear testable hypothesis stated?

-Is the study design appropriate to address the stated objectives?

-Is the population clearly described and appropriate for the hypothesis being tested?

-Is the sample size sufficient to ensure adequate power to address the hypothesis being tested?

-Were correct statistical analysis used to support conclusions?

-Are there concerns about ethical or regulatory requirements being met?

Reviewer #1: Major concerns still arise regarding the interpretation of replicon assays and statistical analysis.

Specific comments:

(1) Figures 3 and 4. Comparison of virus growth of recombinant Asian-I and Cosmopolitan type recombinant dengue virus 2 in Figure 3B presents the means and standard deviation of technical replicates from a single experiment. Similarly, virus titers in the supernatant of Vero cells infected with recombinant virus generated by CPER are presented as means and standard deviation of technical samples. Biological replicates are required to produce data with statistical significance. This is especially relevant as major hypotheses of the study, that is differences in the structural and NS1-2B coding sequences faster may account for the faster growth kinetics of the Cosmopolitan type virus, are based on the data presented in these two panels.

(2) Figure 7. The experimental approach is problematic. Luciferase levels are not significantly different between constructs with the wild type or C709A mutant RdRp until 120 h, and even at this late time point the difference is still not statistically significant for one of the constructs (Gluc-CAAA vs. Gluc-CAAA (C709A), P = 0.1543). Statement made in lines 312-314 does not hold: "At first, luciferase activity is derived simply from the transfected CPER product, but it increases 48 h

after transfection due to the Gluc-containing genome replication supported by non-structural proteins". Also, data in panel B are representative results of three independent experiments. As already mentioned, appropriate statistical analysis requires averaging of biological replicates. Finally, differences in luciferase levels between the recombinant constructs were also noted between RdRp mutants. Does the C709A mutant polymerase retain leaky activity? 

(3) Figure 10. As the figure presents the model of E-M-M-E heterotetramer, the term "envelope tetramer" used across the text should be avoided when referring to this structure. In line with previous comments, molecular modeling is not approapriate to compare binding between envelope proteins of the two types (lines 419-421 "Molecular modeling of the E protein tetramer suggested that the E protein dimer binds more loosely to neighboring E protein dimers in the Cosmopolitan virus than in the Asian-I virus"). 

(4) Figure 11. The authors mix up ionic and hydrogen bonds when referring to the proximity between hydrogen atom in the amino group of pr Lys 52 and oxygen in the carboxyl group pf E Glu 71. Lines 475-477: "Similar to the possible ionic bond between the E protein dimers of the Asian-I virus described above, these results indicated the possibility of the existence of an ionic bond between these residues in the Asian-I type (Fig. 11A, B)". The likely hydrogn bond formed between these two residues is not a ionic bond.

Reviewer #2: This is a revised manuscript. These questions were previously addressed.

Reviewer #3: (No Response)

**Results**

-Does the analysis presented match the analysis plan?

-Are the results clearly and completely presented?

-Are the figures (Tables, Images) of sufficient quality for clarity?

Reviewer #1: Although the results with the different approaches are aligned and point to the envelope protein as a determinant contributing to phenotypic differences between Cosmopolitan and Asian-I types, performing independent biological replicates where noted above can strengthen the analysis.

Reviewer #2: (No Response)

Reviewer #3: (No Response)

**Conclusions**

-Are the conclusions supported by the data presented?

-Are the limitations of analysis clearly described?

-Do the authors discuss how these data can be helpful to advance our understanding of the topic under study?

-Is public health relevance addressed?

Reviewer #1: The authors have included a statement regarding the limitations of the CPER method referring to the rise of unintended mutations during PCR amplification. In the opinion of this reviewer, molecular models are over interpreted and conclusions and discussion referring to these models can be more concise.

Reviewer #2: Yes

Reviewer #3: (No Response)

**Editorial and Data Presentation Modifications?**

Reviewer #1: Figure legends were included in the revised version and references appropriately formatted.

Reviewer #2: (No Response)

Reviewer #3: (No Response)

**Summary and General Comments**

Reviewer #1: In the revised version of the manuscript by Samune and colleagues the authors have responded to the criticisms of the reviewers point by point providing further experimentation and text edits including legends to figures. Overall, major comments were addressed.

Specific comments regarding data analysis and limited value of the replicon assay are made above.

Reviewer #2: This revised manuscript compares two current circulating strains of dengue 2 virus in Thailand and attempt to map the residues responsible for the differential replication and pathogenicity of these strains. The authors have done a reasonable job of responding to and revising their manuscript in response to reviewers’ comments and queries. The manuscript is clearer and the data is more robust in this revision.

Reviewer #3: The authors have adequately addressed my concerns.

PLOS authors have the option to publish the peer review history of their article (what does this mean?). If published, this will include your full peer review and any attached files.

Reviewer #1: No

Reviewer #2: No

Reviewer #3: No

Figure Files:

Data Requirements:

Reproducibility:

References

---

## [Editor Report · Decision Letter 2]

14 Dec 2023

Dear Dr. Shioda,

Thank you very much for submitting your manuscript "Genetic regions affecting the replication and pathogenicity of dengue virus type 2" for consideration at PLOS Neglected Tropical Diseases. As with all papers reviewed by the journal, your manuscript was reviewed by members of the editorial board and by several independent reviewers. The reviewers appreciated the attention to an important topic. Based on the reviews, we are likely to accept this manuscript for publication, providing that you modify the manuscript according to the review recommendations. 

Sincerely,

Kenneth A. Stapleford, Ph.D.

Guest Editor

Abdallah Samy

Section Editor

Editor and reviewers' comments: 

The reviewers raised an important concern (#check below) about your manuscript. Please make sure to address this concern before moving your manuscript forward to an additional revision.

1) There are still issues with Figure 7. No new experiments are needed but modifications to the figure are required. While I understand the need to show "representative" experiments, it is clearer to show the combined averages of the 3 experiments in Figure 7. I believe you have this already in Supp Fig 3. Please make Figure 7B and C the combined averages of the 3 experiments with relevant statistics. Moreover, while you provided P values and statistics in the text, there is nothing in the figure legend I could find regarding which test you used. Please modify the figure legend to reflect the statistics as well.

Figure Files:

Data Requirements:

Reproducibility:

References

---

## [Editor Report · Decision Letter 3]

26 Dec 2023

Dear Dr. Shioda,

We are pleased to inform you that your manuscript 'Genetic regions affecting the replication and pathogenicity of dengue virus type 2' has been provisionally accepted for publication in PLOS Neglected Tropical Diseases.

Best regards,

Kenneth A. Stapleford, Ph.D.

Guest Editor

Abdallah Samy, Ph.D.

Section Editor

---

## [Editor Report · Acceptance letter]

2 Jan 2024

Dear Prof. Shioda,

We are delighted to inform you that your manuscript, "Genetic regions affecting the replication and pathogenicity of dengue virus type 2," has been formally accepted for publication in PLOS Neglected Tropical Diseases.

Best regards,

Shaden Kamhawi

co-Editor-in-Chief

Paul Brindley

co-Editor-in-Chief
